# Spatially resolved multiomics on the neuronal effects induced by spaceflight in mice

Yuvarani Masarapu [1,17], Egle Cekanaviciute [2,17], Zaneta Andrusivova[1,17], Jakub O. Westholm [3], Åsa Björklund [4], Robin Fallegger [5], Pau Badia-i-Mompel [5,6], Valery Boyko [2,7], Shubha Vasisht[8], Amanda Saravia-Butler[9], Samrawit Gebre[2], Enikő Lázár [1,10], Marta Graziano[11], Solène Frapard [1], Robert G. Hinshaw [12], Olaf Bergmann [10,13], Deanne M. Taylor [8,14], Douglas C. Wallace [15], Christer Sylvén[16], Konstantinos Meletis [11], Julio Saez-Rodriguez [5], Jonathan M. Galazka [2], Sylvain V. Costes [2] ✉ & Stefania Giacomello [1] ✉

Impairment of the central nervous system (CNS) poses a significant health risk for astronauts during long-duration space missions. In this study, we employed an innovative approach by integrating single-cell multiomics (transcriptomics and chromatin accessibility) with spatial transcriptomics to elucidate the impact of spaceflight on the mouse brain in female mice. Our comparative analysis between ground control and spaceflight-exposed animals revealed significant alterations in essential brain processes including neurogenesis, synaptogenesis and synaptic transmission, particularly affecting the cortex, hippocampus, striatum and neuroendocrine structures. Additionally, we observed astrocyte activation and signs of immune dysfunction. At the pathway level, some spaceflight-induced changes in the brain exhibit similarities with neurodegenerative disorders, marked by oxidative stress and protein misfolding. Our integrated spatial multiomics approach serves as a stepping stone towards understanding spaceflight-induced CNS impairments at the level of individual brain regions and cell types, and provides a basis for comparison in future spaceflight studies. For broader scientific impact, all datasets from this study are available through an interactive data portal, as well as the National Aeronautics and Space Administration (NASA) Open Science Data Repository (OSDR).

In preparation for long-duration lunar and Mars missions, it is crucial to investigate the health risks to astronauts that are posed by exposure to the space environment[1]. The key physiological impairments include DNA damage and oxidative stress from galactic cosmic radiation[2], bone and muscle damage from loss of gravity[3,4], circadian and sleep dysregulation[5], microbial dysbiosis[6] and tissue and organ degeneration, including cardiovascular and CNS damage[7]. Studies on animal models have shown several spaceflight stressors directly impacting brain molecular mechanisms causing neuroinflammation, neurodegeneration and neurovascular damage[8–11]. Specifically, exposure to simulated space radiation at doses comparable to the ones expected during the planned Mars mission, leads to

neurodegeneration and neuroinflammation in vivo and in tissue and cell models[12], as well as cognitive and behavioral deficits in rodent models. Furthermore, major physiological effects of microgravity in low-Earth orbit, as observed at the International Space Station (ISS), include a redistribution of body fluids from lower towards upper body parts, leading to significant cardiovascular[13] and CNS changes[14]. However, deeper investigations are needed to understand the mechanisms and specificity of these impairments, and identify targets for countermeasure development.

Studies performed directly on astronauts have to be minimally invasive, resulting in limited information on molecular and cellular level outcomes, while spaceflown rodent models allow the range of subcellular to behavioral-level investigation of spaceflight effects. In this study, we used spatial transcriptomics and multiomics to explore spaceflight-induced molecular changes in brain tissue of young adult female BALB/c mice samples obtained during the Rodent Research-3 (RR-3) mission and requested from the NASA Biological Institutional Scientific Collection (NBISC). We combined single-nucleus multiomics (i.e., snMultiomics, RNA and ATAC sequencing) with Spatial Transcriptomics (ST) data to study changes in chromatin accessibility and RNA expression in different cell types and brain regions and to identify key regional and cell type differences in responses to spaceflight.

Single-cell sequencing is a powerful method to obtain information about the cellular composition of a tissue and cellular-level responses to perturbations. However, the applicability of single-cell transcriptomics relies on the accessibility of fresh specimens and the possibility to successfully dissociate the tissue of interest[15]. Since sample preparation in spaceflight studies typically involves in-orbit freezing of the entire carcass to simplify the experiment and minimize crew time, single-nucleus transcriptomics becomes a more feasible method. Using single-nuclei multiomics, we were able to assess both transcriptional profiles and chromatin changes within the same nuclei, thus providing paired data of RNA expression and chromatin accessibility.

While single-nuclei sequencing methods facilitate the study of individual nuclei transcriptomes, they lack spatial context. Therefore, spatial transcriptomics is a powerful addition, enabling the comparison of effects among brain structures and providing transcriptomics information coupled with the original location within tissue sections[16].

Overall, our study is a stepping stone towards the understanding of the molecular changes induced by the space environment at the resolution of individual cell type and brain region. Our findings not only advance the knowledge of the neurological effects of spaceflight, but also provide a critical foundation for future space biology mission designs.

## Results

To identify specific cellular microenvironments affected by spaceflight, we combined the techniques of spatial transcriptomics (ST; 10X Genomics Visium) and single-nucleus multiomics (snMultiomics; gene expression and chromatin accessibility; 10× Genomics Single Cell Multiome ATAC + Gene Expression) on mouse brain. In total, we analyzed three brains from mice euthanized on-board of the International Space Station (ISS; F1, F2, F3) and three brains from ground control mice (G1, G2, G3) that were kept under matched conditions (see Animals in Methods). For each sample, we isolated nuclei from one hemisphere for snMultiomics analysis and cryo-sectioned the other hemisphere for ST analysis with the focus on the hippocampal region (Fig. 1).

### Spaceflight sample quality is suitable for ST and snMultiomics analysis

As a first step, we ensured that the morphological and RNA quality of the samples was suitable for our experimental workflow given that the spaceflown samples had undergone a specific preservation approach[17], which was also used for the corresponding ground control animals (see Animals in Methods). We measured the RNA integrity number (RIN) for each sample and found that it was 9.15 on average (Supplementary Fig. 1A). Furthermore, we performed a tissue optimization experiment confirming that both RNA integrity and tissue morphology was of sufficient quality for ST analysis (see Visium Spatial Gene Expression technology and sequencing in Methods; Supplementary Fig. 1B).

### snMultiomics dissects the effects of spaceflight on different cell types in spaceflown mouse brains

To dissect the alterations induced by spaceflight at the single-nucleus level, we performed a snMultiomics analysis on hemispheres of three spaceflown (F1, F2, F3) mice and two out of three ground controls

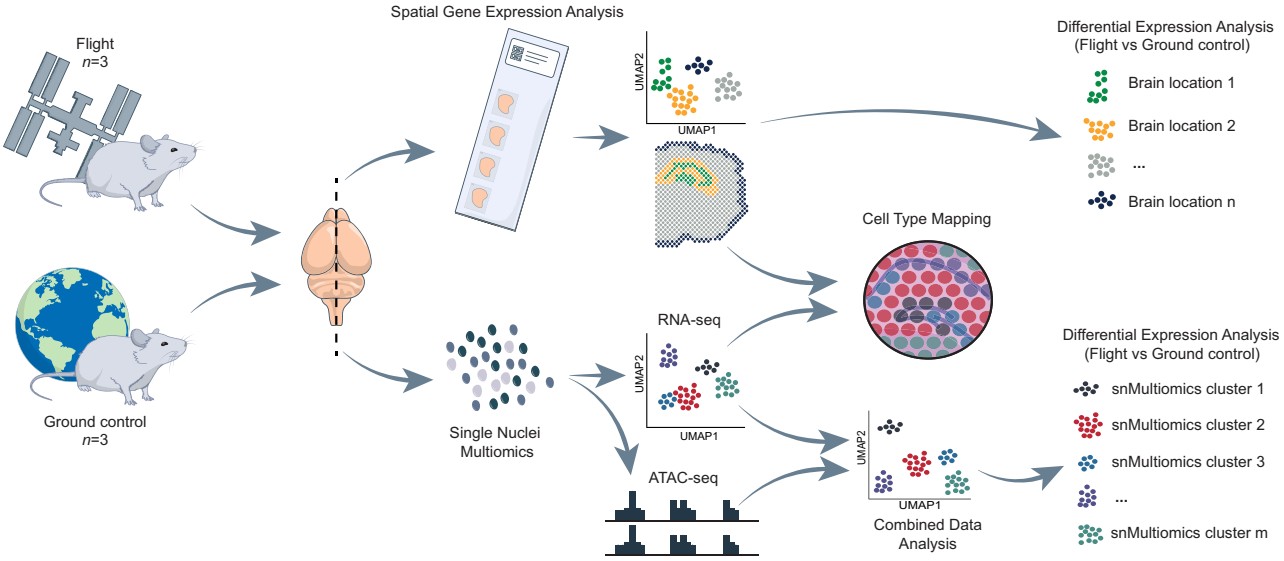

**Fig. 1 | Summary of the project workflow.** Overview of the study workflow where brains from International Space Station (ISS; Flight mice) and ground control mouse groups (Ground control mice) were split into the two hemispheres for Spatial Gene Expression Analysis (Spatial Transcriptomics or ST) and Single Nuclei Multiomics analysis (snMultiomics).

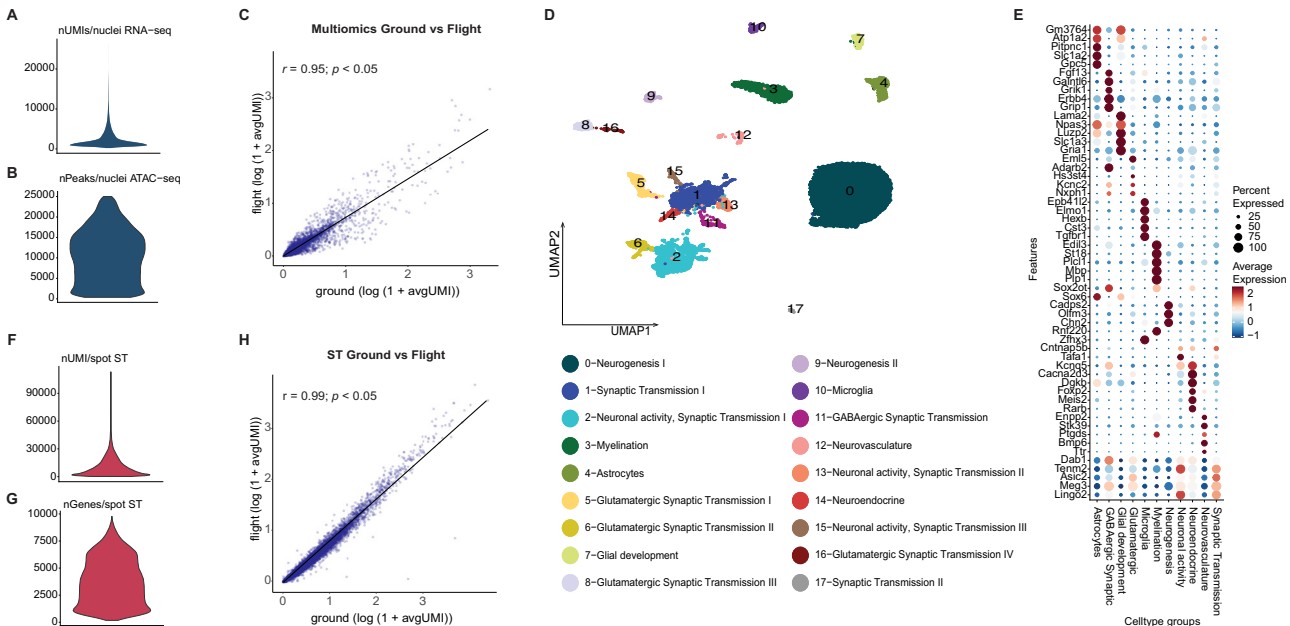

**Fig. 2 | Single-nucleus multiomics analysis of spaceflown mouse brains.**
**A** Distribution of UMIs per nucleus in the entire snRNA-seq dataset. nUMI/nuclei: number of UMIs detected in each nuclei. **B** Distribution of peaks per nucleus in the entire snATAC-seq dataset. nPeaks/nuclei: number of peaks detected per nuclei in the multiomics dataset. **C** Correlation between flight (y-axis) and ground control (x-axis) single nuclei multiomics samples (Pearson's correlation coefficient, $r = 0.95$; $p < 0.05$) shown as a scatter plot. This is a two-sided Pearson correlation test with 95% confidence intervals performed on the average expression ($\log(1 + avgUMI)$). avgUMI: average UMI counts per spot. **D** UMAP of single nuclei multiomics data and cluster annotations. **E** 11 functional multiomics clusters categories represented by their marker genes. **F** Distribution of UMIs per spot for the whole spatial transcriptomics (ST) dataset. nUMI/spot: number of UMIs detected per spot in the ST dataset. **G** Distribution of unique genes per spot for the whole spatial transcriptomics (ST) dataset. nGenes/spot: number of genes detected per spot in the ST dataset. **H** Correlation between flight (y-axis) and ground control (x-axis) ST samples (Pearson's correlation coefficient, $r = 0.99$; $p < 0.05$) shown as a scatter plot. This is a two-sided Pearson correlation test with 95% confidence intervals performed on the average expression ($\log(1 + avgUMI)$). avgUMI: average UMI counts per spot.

(G2, G3), obtaining RNA expression profiles (RNA-seq) and chromatin accessibility (ATAC-seq) information from the same nucleus.

In total, we isolated 21,178 nuclei across the spaceflight and control samples with an average of 3140 unique transcripts (Unique Molecular Identifier or UMI) per nucleus (i.e., from snRNA-seq) and 9217 peaks per nucleus (i.e., from snATAC-seq) (Fig. 2A, B; Supplementary Fig. 1C) and an overall high gene expression correlation between the spaceflight and ground control samples ($r = 0.95$, $p < 0.05$; Fig. 2C). By integrating snRNASeq and snATAC-seq data and performing a joint clustering analysis, we identified 18 snMultiomics clusters (Fig. 2D; Supplementary Fig. 2).

Next, we leveraged previously reported marker genes in the literature (see Gene and cluster annotation in Methods for details) to identify 11 macro categories for the 18 snMultiomics clusters (interchangeably referred to as multiomics clusters in the next sections) according to their functions (Fig. 2E; Supplementary Data 1, 2). The majority of clusters were related to neurogenesis, neuronal activity and synaptic transmission, distinguished by differences in neurotransmitters (GABAergic, glutamatergic, dopaminergic) and based on gene expression patterns, tentatively associated with neuronal locations in hypothalamus, striatum, cortex and hippocampus.

We identified a total of 825 differentially expressed genes (DEGs) between spaceflown and ground control samples across all multiomics clusters (Supplementary Data 3). The majority of these 825 DEGs were involved in neuronal development (multiomics clusters 9, 11), axonal or dendritic outgrowth (multiomics cluster 9), and synaptic transmission (multiomics cluster 4), including specifically GABAergic synaptic transmission (multiomics cluster 11).

Comparison of 825 spaceflight multiomics DEGs to the 629 significant DEGs (Spaceflight vs Ground Control; $p$-value < 0.05) from the bulk RNAseq data of the same mice brains from the same NASA mission (RR-3), indicated 11 shared genes ($p$-value = 0.01582549, hypergeometric distribution test; see Gene overlap test in Methods; Supplementary Data 4). Out of these 11 overlapping genes, only 2 genes (Gabra6, and Kctd16) showed the same directional change in both the datasets indicating that the majority of spaceflight effects are cell type-specific and emphasizing the need for cell-specific analysis of central nervous system responses to spaceflight.

We also compared these 825 spaceflight DEGs with spaceflight DEGs reported in a total of 11 other datasets processed by NASA OSDR including mass spectrometry and RNA-seq data collected from different organs of BALB/c and C57BL/6J mice strains. This comparison revealed a total of 461 overlapping DEGs ($p$-value < 0.05) across all the 11 datasets combined (refer to Supplementary Data 5 for a detailed list of overlapping genes and the resulting $p$-value from the hypergeometric distribution test performed for each dataset).

## ST provides a high-resolution mapping of gene expression changes to mouse brain regions

To investigate spaceflight-induced CNS alterations at a spatial level, we performed ST analysis on the other brain hemispheres from 3 flight (F1, F2, F3) and 3 ground control mice (G1, G2, G3). We collected two coronal sections from each brain hemisphere containing hippocampus, somatosensory cortex, striatum, amygdala and corpus callosum.

In total, we captured 14,630 genes across 29,770 spots after filtering and detected 10,884 UMIs/spot and 3755 genes/spot on average (Fig. 2F, G; Supplementary Fig. 3A, B) and found a high overall gene expression correlation between spaceflight and ground control tissue sections ($r = 0.99$, $p < 0.05$; Fig. 2H). Unsupervised clustering analysis of spot information identified 18 distinct spatial clusters (further referred as ST clusters) (Fig. 3A, B; Supplementary Data 6), which

presented a clear separation between the cortical top (ST cluster 1) and bottom layers (ST cluster 9), as well as other major structures, including hippocampus (with separation of CA1, CA3, and dentate gyrus in ST clusters 10, 8 and 11 respectively), thalamus (ST cluster 5), striatum (ST clusters 0, 14), hypothalamus (ST cluster 2), pituitary (anterior and posterior; ST cluster 2), corpus callosum (ST cluster 12) and cerebral peduncles (ST cluster 4) (Fig. 3C). Key functions of the markers (Supplementary Data 7) that were shared by numerous ST clusters include neurogenesis, neuronal development, axonal growth and synaptogenesis, indicating that ST cluster analysis is dominated by neuronal gene expression.

Next, we investigated how spaceflight influences gene expression at the spatial level and identified a total of 4057 DEGs in 7 out of 18 ST clusters (Supplementary Data 8). The majority of DEGs were involved in neuronal development, synaptogenesis and synaptic plasticity, and neurodegeneration, including 21 DEGs in hippocampal CA3 neurons. The most pronounced change in gene expression due to spaceflight was observed in cortical neurons (bottom layers; ST cluster 9) which showed 3208 DEGs (1808 upregulated, and 1400 downregulated) with similar functions related to neuronal development and synaptic transmission in somatosensory, motor and visual cortex. Consensus pathway analysis[18] highlighted neurodegeneration-associated pathways in cortical neurons (bottom layers; ST cluster 9) including protein misfolding and abnormal protein clearance, indicating potential similarities with neurodegenerative diseases characterized by protein misfolding and accumulation, such as Parkinson's disease[19,20] (Fig. 3D).

### Integration of multiomics and ST datasets for spatial and cell type-level effects of spaceflight

To infer the spatial distribution of the clusters identified by multiomics, we performed spot deconvolution analysis on matching ST dataset using Stereoscope[21] (which corrects for biases arising from different experimental techniques before calculating celltype proportions probabilities) (Fig. 3E; refer to Supplementary Figs. 4–6 for detailed visualizations of multiomics cluster proportions in ST dataset). The deconvolution analysis revealed similarities based on the assigned functional annotations between several multiomics and spatial data clusters, for instance, synaptic transmission (multiomics cluster 1 matched with ST clusters 0 and 2), myelination (multiomics cluster 3 matched ST clusters 4 and 12), and neuronal activity (multiomics cluster 15 matched ST cluster 5) (Fig. 3F; Supplementary Figs. 7, 8; Supplementary Data 9). This comparative analysis suggested the effects of spaceflight on synaptic transmission specifically in cortex (including both neurons and astrocytes, as revealed by snRNA-seq data that allowed cell type separation) and on dopaminergic neuron development specifically in striatum (Supplementary Data 9).

### Ligand-receptor interaction analysis suggests spaceflight-mediated effects on astrocyte functions

To assess the effects of spaceflight on the cell-cell interaction level, we performed a ligand-receptor analysis on two multiomics clusters that showed among highest number of differentially expressed genes in response to spaceflight, i.e., multiomics clusters 4 (Astrocytes), and 11 (GABAergic Synaptic Transmission). We found 4 significantly upregulated interactions (Fig. 4A), including adhesion molecule pairs, EGFR (epidermal growth factor receptor) pairs, and VEGFA (vascular endothelial growth factor). These ligand-receptor interactions have previously been shown to be involved in cellular development in the CNS. EGFR[22], is involved in neuronal development, including axonal outgrowth. Meanwhile, VEGFA[23,24] primarily regulates angiogenesis though it can also play a role in hippocampal neurogenesis, and astrocyte-produced VEGFA has previously been demonstrated to regulate neuronal NMDA receptor activity[23–25]. Interestingly, we found that spaceflight widely increased VEGFA_GRIN28 interactions between multiomics cluster pairs related to astrocytes and synaptic

transmission, i.e., 4-11 (Astrocytes-GABAergic Synaptic Transmission). No ligand-receptor interactions in these clusters were significantly downregulated.

We also extended the ligand-receptor analysis to the ST dataset using SpatialDM[26]. We applied SpatialDM on each ST brain section to identify spatially co-expressed LR pairs and found a total of 1260 LR pairs (Supplementary Fig. 9; refer to Supplementary Data 10 for a detailed list of LR pairs with corresponding z-scores across each ST section). Differential testing between the two conditions (flight and ground control) for the observed 1260 LR pairs revealed a total of 134 differential LR pairs (differential $p$-value < 0.1; Supplementary Data 11).

### Motif analysis shows spaceflight effects on transcription factor (TF) activity

To investigate the effects of spaceflight on transcription factors (TFs), we performed motif analysis on snATAC-seq peaks from the single nucleus multiomics data, which revealed spaceflight-mediated differences in TF activity in several multiomics clusters (Supplementary Data 12), especially 4 (Astrocytes), and 11 (GABAergic Synaptic Transmission).

Spaceflight was associated with reduced accessibility of motifs Zic1, Zic2 and Atoh1 in multiomics clusters 4 (Astrocytes)[27,28] (Fig. 4B). Meanwhile, increased accessibility of motifs Pou5f1 and Sox2 in multiomics cluster 11 (GABAergic Synaptic Transmission) might indicate reduced neuronal differentiation in spaceflight[29–31] (Fig. 4C). In addition to neuronal effects, motifs Pparg, Rxra and Nr2f6, which collectively inhibit immune responses, showed decreased accessibility in telencephalon interneurons (multiomics cluster 11), suggesting increased inflammatory responses in space[32–34], and possible circadian dysregulation[35–39].

### Spatial gene expression patterns show several changes in signaling pathways occurring in spaceflight brain samples

Local environments of cell types may affect the functional responses to spaceflight represented by changes in signaling pathways. We compared key signaling pathways in adjacent locations based on the spatially-resolved cell type deconvolution results from Stereoscope analyzed using the Multiview intercellular SpaTial modeling framework (MISTy)[40]. This tool allowed us to investigate the relationships between cell type proportions in each ST spot and activities of 14 pathways inferred by decoupler-py and PROGENy[41,42]. Specifically, the MISTy models predict cell type abundance in a spot based on an intraview (features in the same spot) and paraview (weighted sum of the features in the neighboring spots; weights decreasing with distance). Either cell type abundances or pathway activities were selected as "features" for the model, and a separate model was built for each sample and cell type. To analyze the effects of spaceflight, the models were subsequently aggregated into flight and ground control groups.

Based on cell type abundances, we did not find any significant changes in cell type colocalization (which would occur during tissue restructuring or lesion formation) between flight and ground controls, similar to our previous finding of no significant changes in cell type abundance in deconvolution results (Supplementary Figs. 7 and 8).

In contrast, changes in signaling pathways were associated with individual cell types. Cell abundance in neurovasculature (multiomics cluster 12) colocalized with decreased MAPK signaling in spaceflight (Fig. 4D). Similarly, signaling changes in local neighborhood (MISTy paraview) of several other cell types were found in spaceflight samples (Fig. 4E): (1) less negative correlation of EGFR signaling and glutamatergic neuronal cells; (2) more negative correlation of MAPK and cholinergic, monoaminergic and peptidergic neurons; (3) increased TGFbeta signaling in the vicinity of GABAergic interneurons; (4) reduced WNT signaling in class II glutamatergic neurons.

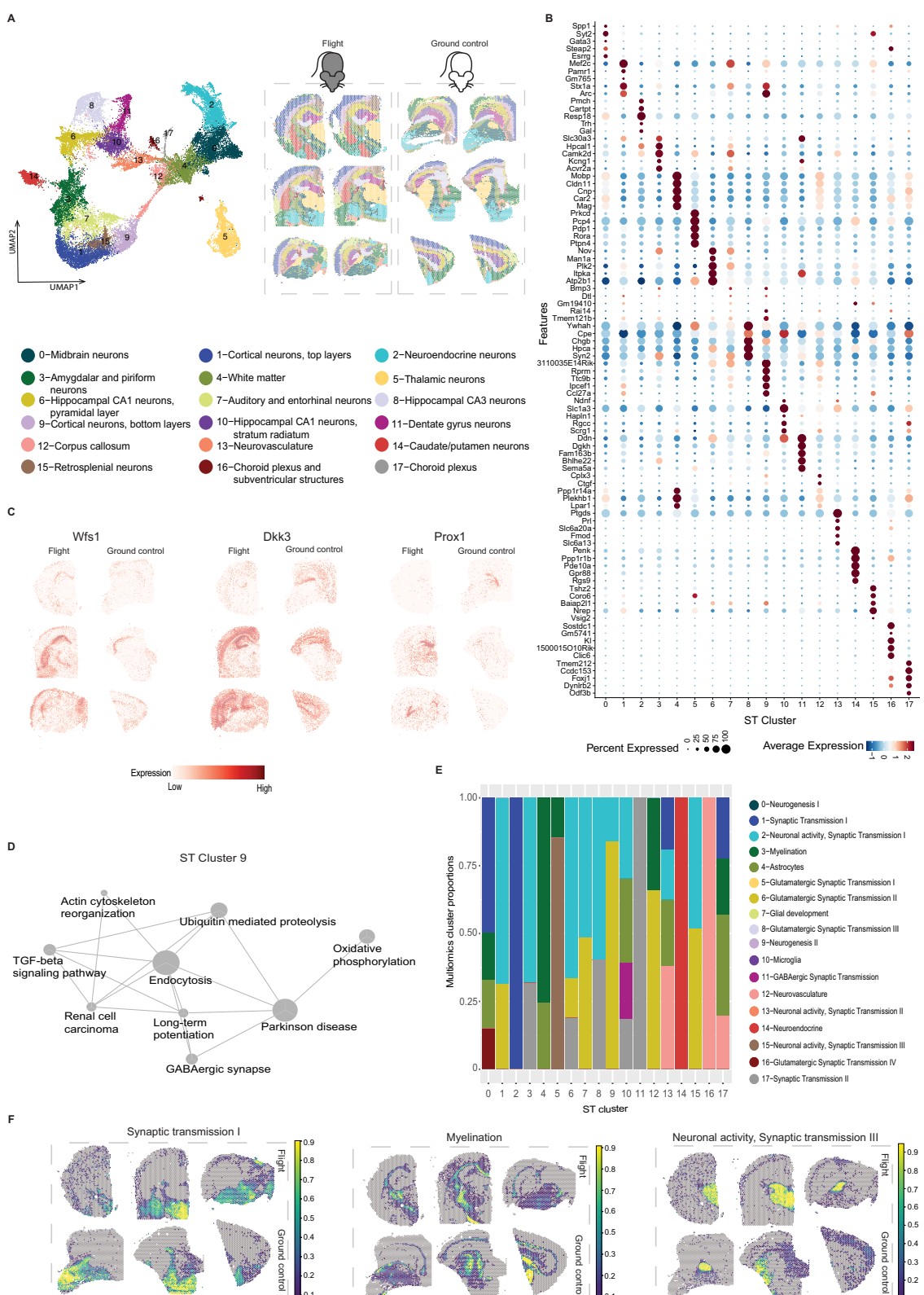

**Fig. 3 | Spatial Transcriptomics datasets, cell type deconvolution and pathway analysis of ST data. A** Clustering of spatial transcriptomics data, cluster annotations and spatial location of clusters visualized on flight and ground control mouse brain sections. **B** Marker genes for each ST cluster visualized as dotplot. **C** Spatial distribution of 3 genes (Wfs1 for CA1 region of hippocampus, Dkk3 for CA1 and CA3 hippocampal region and Prox1 for Dentate gyrus) in three flight (left column) and three ground control (right column) ST sections. **D** Significantly different pathways ($p < 0.05$) between flight and ground control in ST cluster 9 (Cortical neurons,

bottom layers). **E** Visualization of number of clusters identified by single-nuclei multiomics and their proportions in each ST cluster (x-axis; 0–17). Only multiomics clusters with higher proportions (>10%) are displayed in the barplot. **F** Cell type proportions mapped to spatial coordinates on three ground control (top row) and three flight (bottom row) mouse brain sections (Synaptic transmission I or multiomics cluster 1; Myelination or multiomics cluster 3; Neuronal activity, Synaptic transmission III or multiomics cluster 15).

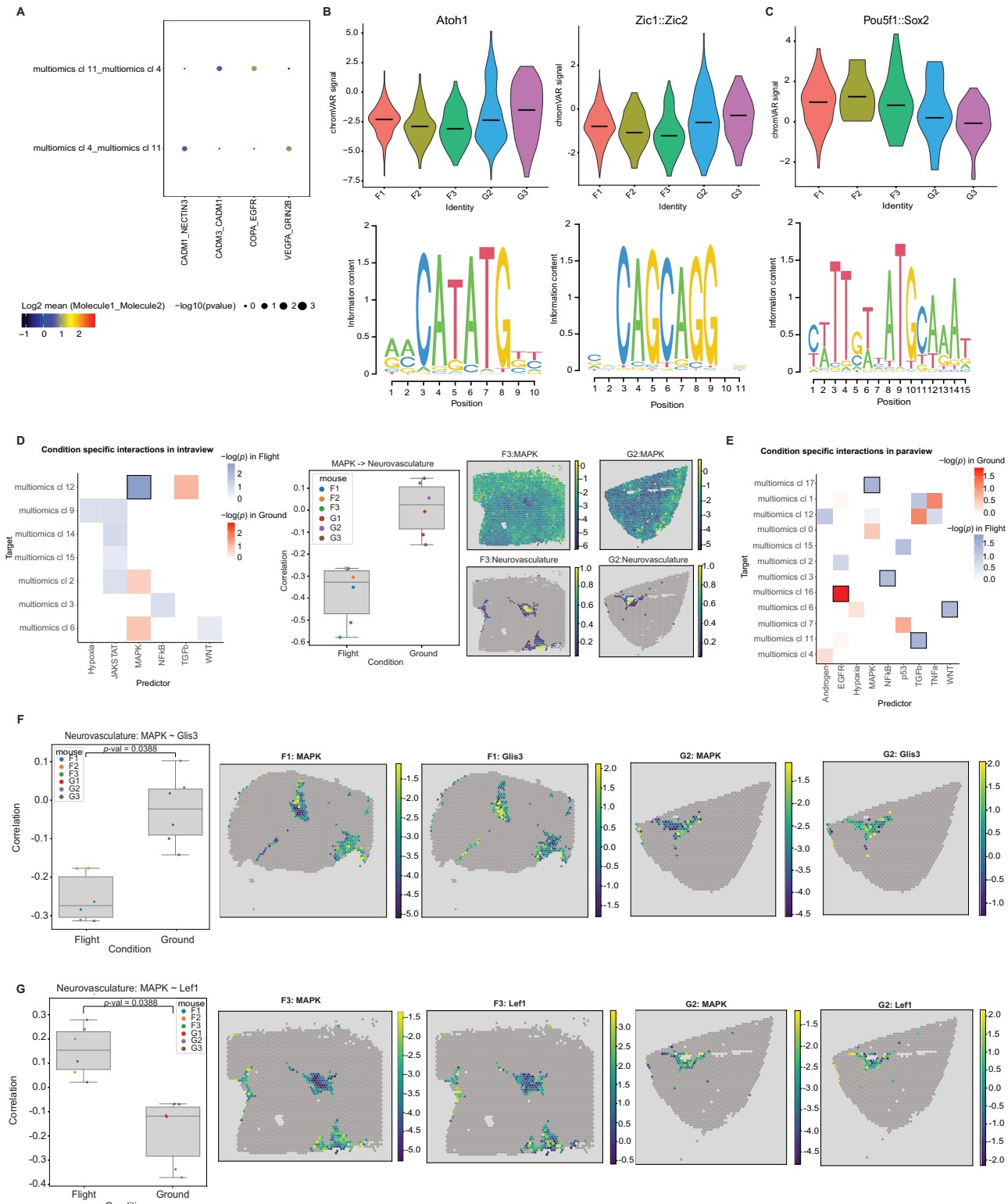

To assess the downstream effects of these changes, we built a tissue-specific gene regulatory network (GRN) from the multiomics data using CellOracle[43] and used it to predict TF activities in spatial data and computed the Pearson correlation between TF and signaling activities for the dysregulated pathways in spots containing the cell types identified above. The network suggested that the decrease in MAPK signaling in spaceflight increases activity of the transcription factor Glis3 and reduces Lef1 in neurovasculature, respectively (Fig. 4F, G).

## Metabolic gene enrichment analysis shows decreased gene expression in spaceflight mice

Gene Set Enrichment Analysis (GSEA) on the ST data using metabolic pathways indicated spaceflight-mediated inhibition of the oxidative phosphorylation pathway, especially Complex I signaling (Fig. 5A, Supplementary Data 13), as well as pathways related to glycolysis/gluconeogenesis (Supplementary Fig. 10), fructose and mannose metabolism (Supplementary Fig. 11) and arachidonic acid metabolism (Fig. 5B). Analysis of multiomics data was consistent with spaceflight-

**Fig. 4 | Ligand-receptor interactions, motif accessibility, and signaling pathways affected by spaceflight. A** Dotplot showing the differentially expressed ligand receptor pairs found by CellPhoneDB between two interacting multiomics clusters (4 and 11) which are affected by spaceflight. These clusters showed the largest number of spaceflight DEGs, and four LR pairs were found significantly upregulated in these interactions. The null distribution of the mean expression of the LR pairs was estimated by employing a random permutation approach. The mean expression of the interacting LR molecule pairs are indicated by the dot colors and the dot sizes represent the $p$-values which refers to the enrichment of the LR pair in the interacting multiomics clusters. Scales for both dot size and color are presented below the plot. **B** Accessibility differences for motifs Atoh1, Zic1, and Zic2 in multiomics cluster 4 of flight mice and ground control mice. Spaceflight results in reduced accessibility of these motifs in flight samples. Two-sided Chi-square test statistic was used for differential testing with FDR correction (fdr <0.05). **C** Accessibility differences for motifs Pou5f1, and Sox2 in multiomics cluster 11 of flight and ground control mice. Spaceflight results in increased accessibility of these motifs in flight samples. Effects of spaceflight shown by increased accessibility of these motifs in flight samples. Two-sided Chi-square test statistic was used for differential testing with FDR correction (fdr <0.05). **D** (left) adjusted $p$-value of differential interactions found by MISTy in intraview (cell type and pathway activity colocalization) occuring only in flight (blue; $n = 3$ individual ST flight mouse samples) or in controls (red; $n = 3$ individual ST ground control mouse samples), tiles with black border identify statistically significant changes, (middle) correlation of MAPK pathway activity and Neurovasculature abundance, and mapped on Visium slide for two samples (right). Two-sided Student's $t$ tests with Benjamini–Hochberg multiple testing correction was used to determine the differential interactions. **E** adjusted $p$-value of differential interactions found by MISTy in paraview (cell type and pathway activity in local neighborhood) occuring only in flight (blue; $n = 3$ individual ST flight mouse samples) or in controls (red; $n = 3$ individual ST ground control mouse samples), tiles with black border identify statistically significant changes. Two-sided Student's $t$ tests with Benjamini–Hochberg multiple testing correction was used to determine the differential interactions. **F** Pearson correlation of Glis3 activity (left) containing vascular endothelial cells and MAPK activity ($n = 6$ individual ST mouse samples, 3 flight, 3 ground controls), and their respective activities in Visium slides (4 plots on the right). Two-sided Student's $t$-tests with Benjamini–Hochberg multiple testing correction was used to determine the changes in correlation. **G** Pearson correlation of Lef1 activity (left) within spots containing vascular endothelial cells and MAPK activity, and their respective activities in Visium slides (4 plots on the right). Two-sided Student's $t$ tests with Benjamini–Hochberg multiple testing correction was used to determine the changes in correlation. multiomics cl: multiomics cluster. The boxplots in **D**, **F**, and **G** show the median as a central line, the box boundaries denote the first and third quartiles and the whiskers extend to the most extreme point in the range within 1.5 times the interquartile range from the box.

mediated reduction in these pathways together with fatty acid synthesis (Fig. 5C; Supplementary Data 14). Deficits in glycolysis and oxidative phosphorylation are consistent with previously reported mitochondrial impairments caused by spaceflight[44], while, arachidonic acid is primarily produced by astrocytes and suggests astrocyte dysfunction as a potential target for future spaceflight CNS studies.

### Validation of differential gene expression

In order to validate our findings on the spaceflight affected processes in mouse brain, we performed single molecule Fluorescence In situ Hybridization (smFISH) using the RNAscope technology for two genes of interest (Adcy1 and Gpc5) in five brain sections: 3 flights, 2 ground controls (Supplementary Fig. 12) from a comparative set of mice (see Methods). We observed significant upregulation in the expression of both genes in spaceflight samples, confirming our findings from the ST data and multiomics data analysis (Supplementary Data 3 and 8, Supplementary Fig. 13A–C). Adcy1 was particularly upregulated in the hippocampus and associated with changes in neuronal activity (ST clusters 8, 11), while Gpc5 was upregulated in astrocytes (multiomics cluster 4).

## Discussion

In this study, we used brain samples from ground control and spaceflown female mice from the RR-3 mission to understand the effects of the space environment at high resolution: at the level of individual brain cell types and brain regions. The main alterations in gene expression induced by spaceflight included changes in synaptogenesis and neuronal development, as well as neurodegeneration and inflammation, similar to previously reported tissue and behavioral-level effects of simulated spaceflight and simulated deep space radiation in mice[12]. In addition, multiple gene expression changes mapped to pathways associated with circadian rhythm and mitochondrial functions. This study applies spatial transcriptomics and single-cell multiomics to study the effects of spaceflight on the CNS which can serve as a stepping stone towards comparative investigation of samples from other space missions and the outcomes of live animal return and reacclimation to Earth conditions.

We demonstrate spatial transcriptomics and single nuclei multiomics datasets to be highly complementary. For example, single-nucleus resolution permitted a focus on glial cells including astrocytes and microglia, which were not distinguishable in neuron-dominated ST datasets, due to the lower resolution of the method. Similarly, our snATAC-seq analysis in transcription factor binding sites indicated

immune dysfunction[34], while immune cells were too few to be distinguishable in ST. On the other hand, combining spatial and gene expression data revealed region-specific effects of spaceflight[32,45]. Furthermore, analysis of ligand-receptor interactions revealed interactions between different cell types, with the focus on neurodevelopment and astrocyte functions.

In our study, multiple lines of evidence converged on spaceflight-mediated disruption of neurogenesis, neuronal development and synaptogenesis, including dopaminergic synapse formation in the striatum. Similar outcomes of increased pluripotency and reduced cell differentiation have previously been reported in spaceflown mouse bone samples[46], in spaceflown human stem cell models[47], and in irradiated mouse models[48], indicating that it might be a systemic impairment that has now been identified to include the CNS. In addition, neurogenesis has been shown to be affected in a spaceflight analog study that combined simulated microgravity and low dose rate photon irradiation in mice[10]. Notably, synaptogenesis is particularly sensitive to simulated galactic cosmic radiation components[49], indicating a potential additional risk for deep space missions. Finally, our observations in the striatal gene expression changes and transcription factor accessibility are worth validating in future spaceflight experiments, since they might result in disrupted dopaminergic signaling.

Spaceflight-associated changes in circadian gene expression are consistent with circadian disruption as one of the major biological responses to spaceflight[50], though it has been studied comparatively little so far, especially in animal models[51]. Molecular and cellular CNS correlation of circadian gene expression dysregulation in spaceflight remains to be investigated in greater detail.

Our results suggest potential similarities between spaceflight and some terrestrial CNS conditions. Changes in gene expression associated with disrupted synaptogenesis and neurogenesis, oxidative stress and inflammation collectively resemble neurodegeneration that is associated with natural aging. Thus, in future studies anti-aging and neuroprotective therapeutics, such as antioxidants[52] or mitochondria-targeting pharmaceuticals[53,54], might be repurposed as spaceflight countermeasures, or vice versa, spaceflight might be used as a model for accelerated aging[55]. We have recently begun computational studies on identifying terrestrial disorder analogs to alterations induced by spaceflight, though they have not yet included CNS effects, which will require comparing the data from multiple spaceflight experiments[56].

Our study presents two main limitations. The first one is related to statistical robustness. Specifically, since the input material consists of spaceflight legacy samples collected without envisioning a spatial

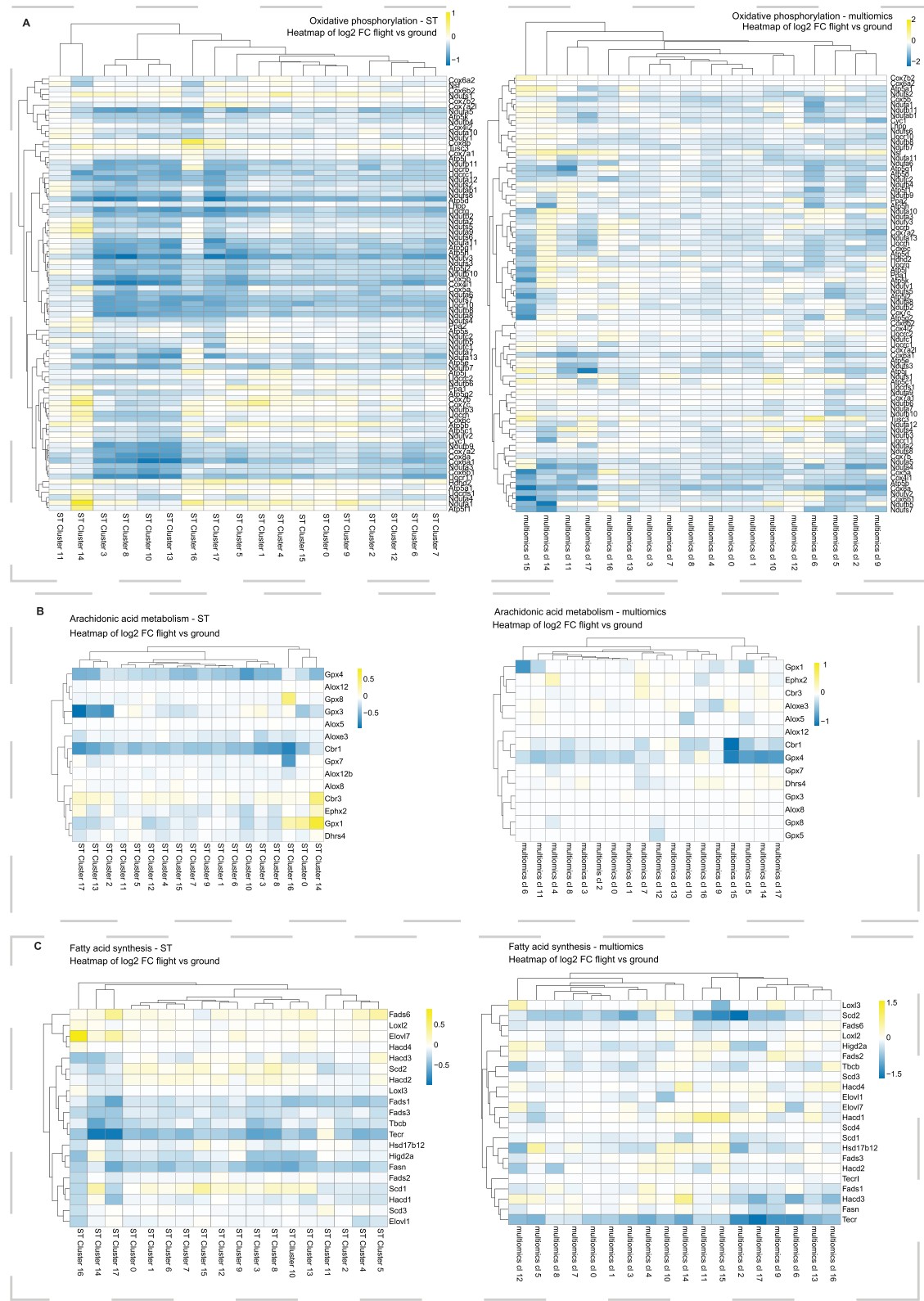

**Fig. 5 | Metabolic gene set enrichment analysis. A** Heatmap showing fold change differences (log2FC) between flight and ground control samples in oxidative phosphorylation pathway in both ST and multiomics datasets. There is a spaceflight-mediated inhibition seen for this pathway that is consistent across the two datasets. Two-sided Wilcoxon's rank-sum test was done with FDR adjustment. **B** Heatmap showing fold change differences (log2FC) between flight and ground control samples in Arachidonic acid metabolism pathway in both ST and

multiomics datasets. There is a deficit for this pathway seen in spaceflight samples in both the datasets. Two-sided Wilcoxon's rank-sum test was done with FDR adjustment. **C** Heatmap showing fold change differences (log2FC) between flight and ground control samples in Fatty acid synthesis pathway in both ST and multiomics datasets. There is a spaceflight-mediated reduction observed for this pathway in both the modalities. Two-sided Wilcoxon's rank-sum test was done with FDR adjustment. multiomics cl: multiomics cluster.

transcriptomics analysis, the resulting brain morphology is not fully even across samples. This limits the interpretability of the differences between spaceflight and ground control conditions to only a subset of spatial and multiomics clusters. Therefore, it will be important to validate the results of the RR-3 mission by comparing them to the neuronal effects of spaceflight in other mouse studies at similarly high resolution to increase statistical robustness. Moreover, it will be meaningful to expand the experimental design by including samples from animals that underwent a period of reacclimation to ground conditions after spaceflight. By doing so, it will be possible to quantify the extent to which spaceflight-mediated impairments persist after landing, especially once model organisms begin to age. This will allow the investigation of key aspects of astronaut health risks that have not been included yet.

The second limitation of our study relates to the sex of the analyzed animals. The majority of mouse studies in spaceflight have been performed on females to facilitate group housing, while astronaut cohorts include both male and female astronauts. Thus, it will be necessary to compare male and female animals to address the sexual dimorphism of CNS health risks. In order to further extrapolate animal models to human outcomes, human and mouse CNS organ model systems will have to be flown as payloads. The first human neurovascular models have already been investigated on the ISS[57,58], though their results have not yet been published, while analyzing the effects of simulated space radiation on human neurovascular models[59] highlighted the importance of investigating astrocyte functions in missions beyond low-Earth orbit. In preparation for these missions, in which crew time will be severely limited, it will be essential to develop automated, long-lasting CNS models and compare their responses to model organisms.

Overall, our study serves as a preliminary indication to the potential risks that the brain is exposed to in spaceflight missions and more studies with larger sample sizes will be required to confirm our findings. To encourage the scientific community to use our datasets for generating new discoveries, all our data is openly available through NASA GeneLab and can be explored by a Shiny app (https://giacomellolabst.shinyapps.io/rr3-brain-shiny/).

In summary, in this study, we observed that spaceflight altered the expression of genes associated with neurogenesis, synaptogenesis and neuronal development in multiple regions of the brain, especially striatum, as well as oxidative stress and neuroinflammation collectively resembling neurological aging and neurodegeneration. Future studies are planned to increase statistical robustness, focus on neuroimmune outcomes and investigate the persistence of spaceflight effects after landing and re-acclimation to Earth environment. In addition, our findings illustrate the benefits of combined spatial transcriptomics and single-cell analysis in revealing detailed biological alterations and suggest them as highly relevant techniques for future space biology research. Our shared data, freely available in NASA's OSDR database[60], aims to deepen our collective understanding of the impact of spaceflight on biological systems.

## Methods

### Ethical approval
The study followed recommendations in the Guide for the Care and Use of Laboratory Animals and the protocol (CAS-15-001-Y1) was approved by the NASA Flight Institutional Animal Care and Use Committee (IACUC) for both flight (housed at the NASA Ames Research Center) and ground control (housed at the Kennedy Space Center) mice.

### Animals
A cohort of 12-week-old female BALB/c mice were flown to the ISS and housed in the Rodent Habitat for 39–42 days. Male mice were not used in this study because of their behavioral impairments when co-housed

in a stressful environment. Recent studies have also shown that the ISS differences across individual male mice are greater than those across female mice[61], indicating negligible effects of any hormonal changes during estrus cycle in female mice. Mice of similar age, sex and strain were used as ground controls housed in identical hardware (Rodent Habitat AEM-X, identifier 1379) and matching ISS environmental conditions[62], including but not limited to cage type, light cycle (standard 12:12 light/dark cycle for both ground control and flight mice groups), food (Nutrient Upgraded Rodent Food Bar (NuRFB)), temperature (23.36 °C, and 21.37 °C on average for ground control and flight mice, respectively), humidity (41.86% and 41.53% on average for ground control and flight mice, respectively) and $CO_2$ concentration (3664.89 ppm mean $CO_2$ for ground controls, and 3711.97 ppm mean $CO_2$ for flight mice). Detailed information on environmental telemetry data including temperature, humidity, CO2 levels, and cabin radiation exposure can be found on NASA's Environmental Data Application (EDA) by clicking on the publicly accessible link https://visualization.osdr.nasa.gov/eda/ and selecting 'RR-3' from the drop-down menu in the Mission Dashboard on the right-hand panel. Additionally, all information on RR-3 mission payload is publicly available on NASA OSDR link https://osdr.nasa.gov/bio/repo/data/payloads/RR-3.

At the end of the mission, spaceflight and ground controls were euthanized (using euthanasia chemical ketamine/xylazine/acepromazine) and whole carcasses were stored at −80 °C until the dissection on earth. Mouse carcasses were removed from −80 °C storage and thawed at room temperature for 15–20 min for dissection. Each carcass dissection time was no longer than 1 h from collection of the first tissue. Brain tissues were harvested and placed into 2 ml Eppendorf tubes for snap freezing in liquid nitrogen, then stored in −80 °C freezer for sample processing.

### RNA extraction
RNA was extracted from tissue sections using the RNeasy Mini kit (Qiagen). RNA integrity was measured by using the Agilent Bioanalyzer. All samples had RIN values above 8.

### Sample preparation
Six brains in total, three from each group, were used in this study. Hemispheres of each of the brain samples were split. One hemisphere from each sample was embedded in Tissue-Tek O.C.T. and cryosectioned in the hippocampus area at 10 μm thickness. Sections were placed on Visium Gene Expression arrays, Superfrost glass slides or into Eppendorf tubes. Second hemisphere of each brain was used for nuclei isolation. Samples were stored in −80 °C before processing.

### Visium Spatial Gene Expression technology and sequencing
First, 10× Genomics Spatial Gene Expression Tissue Optimization protocol was performed on 8 consecutive mouse brain sections to determine the optimal permeabilization time. One tissue section was used as a negative control. The experiment was performed according to the manufacturer's instructions (User Guide, CG000283 Rev B).

Spatial gene expression libraries were generated using the Visium Gene Expression kit (10x Genomics). Brain hemispheres were cryosectioned to reach the hippocampus area. Two consecutive sections of hippocampus from the ground control cohort and two consecutive sections of hippocampus from the flight cohort were placed on Visium glass slides. Consecutive sections were considered technical replicates. In total, 12 Visium libraries were prepared following the manufacturer's protocol (User Guide, CG000239 Rev F). Twelve libraries were sequenced by using an Illumina Novaseq platform, while four were sequenced on an Illumina NextSeq500 platform and sequenced by using a Illumina Novaseq platform. Length of read 1 was 28 bp and read 2 was 120 bp long.

## Chromium single-cell Multiome ATAC + Gene Expression technology and sequencing

**Nuclei isolation.** One hemisphere of 4 brains used for Spatial Transcriptomics was utilized for nuclei isolation. Brain tissue was placed into a tube together with lysis buffer (10 mM Tris-HCl, 10 nM NaCl, 3 nM MgCl$_2$, 0.1% Igepal CA-630, 1 mM DTT, 1U/µl RNAseOUT), homogenized by pestle homogenization in eppendorf tube and incubated on ice for 5 min. The nuclei were extracted by following 10XGenomic protocol for Single Cell Multiome ATAC + Gene Expression Sequencing (User Guide, CG000375 Rev B). Each sample was filtered through a 40 µm cell strainer prior to FACS sorting. Sorted and permeabilized nuclei were pelleted at 500 × $g$ for 5 min at 4 °C, counted and used for Chromium Single Cell Multiome ATAC + Gene Expression library preparation. One hemisphere from the ground control G1 sample was used to optimize the nuclei isolation protocol.

**Cell sorting.** The stained cell nuclei suspension samples were then analyzed and sorted utilizing a BD Bioscience Influx flow cytometer using an 86 µm nozzle. Flow cytometry analyses and sorting were carried out by the following gating strategy: nuclei:singlets:7-AAD positive events. The nuclei sub-population was characterized in the Side Scatter-Forward Scatter plot by back-gating from the 7-AAD (ex. 488 nm, em. 692 nm), singlets were gated in the Side Scatter – Pulse width plot and finally the nuclei were gated in the Side Scatter – 7-AAD plot. The nuclei were collected in a BSA coated Eppendorf tube.

**Single nuclei samples.** Isolated and sorted nuclei were used to generate single nuclei gene expression and ATAC-seq libraries according to the manufacturer's protocol using Chromium Single Cell Multiome ATAC + Gene Expression kit (User Guide, CG000338 Rev E). Finished Gene Expression libraries were sequenced by using an Illumina Next-Seq2000 platform with lengths of read 1 and read 2, 28 bp and 120 bp long respectively. Finished ATAC-seq libraries were sequenced by using an Illumina NextSeq2000 platform. Length of both read 1 and read 2 was 50 bp long.

## Pre-processing and Clustering of Spatial Transcriptomics (ST) and multiomics Data

**ST data generation.** Sequenced Visium libraries were processed using Space Ranger software (version 1.0.0, 10X Genomics). Reads were aligned to the Space Ranger built-in mouse reference genome (mm10-3.0.0) and count matrices were generated using these along with the Hematoxylin and Eosin (H&E) images.

## ST data analysis

ST data analysis was performed jointly in R (v4.1.1) using the Seurat package (v4.0.1) and STUtility (v0.1.0). InputFromTable() was used to generate a Seurat object from the count matrices with specific filtering parameters (min.spot.feature.count = 100, min.gene.count = 100, min.gene.spots = 10, and min.spot.count = 100). The count matrices were enriched for protein-coding and lincRNA genes and then filtered for MALAT1, ribosomal and mitochondrial genes. Next, all spots containing less than 200 genes were removed. Technical variability within data was reduced with SCTransform (SCTransform() with default settings; SelectIntegrationFeatures() with nfeatures = 3000; PrepSCTIntegration() with anchor.features assigned to genes returned from the SelectIntegrationFeatures()) and Harmony (RunHarmony() with group.vars = 'section.name' addressing for batches associated with different Visium slide capture areas) packages. Principal Component Analysis (PCA) was used for selection of significant components (RunPCA() with default 50 PCs computed; 35 selected for the rest of the analysis). FindClusters() with resolution = 0.35 and dims = 1:35 was used to identify ST clusters and then Uniform Manifold Approximation and Projection (UMAP) was used to visualize the same in 2D space.

## Multiomics data generation

The sequenced multiomics (snRNA-seq, and snATAC-seq) libraries were processed using CellRanger ARC (v2) software from 10X Genomics with default settings. Reads were mapped to the mouse genome (mm10 reference 2020-A from 10X Genomics).

## Multiomics data analysis

Data analysis for the snRNA-seq and snATAC-seq data was performed in R (v4.1.1) using the packages Seurat (v4.1.0) and Signac (v1.6.0). In total, extracted single nuclei from 5 brain hemispheres were processed, resulting in 21,178 nuclei used in the analysis. The chromatin assay was generated using CreateChromatinAssay() from the Signac package (min.cells = 10). The annotation data for this was obtained using GetGRangesFromEnsDb() (ensdb=EnsDb.Mmusculus.v79). The snRNA-seq assay was generated using Read10X_h5() available in the Seurat package. MALAT1, ribosomal and mitochondrial genes were removed and count matrix was further filtered on number of reads, genes and peaks, as well as fraction ATAC-seq reads in peaks, and enrichment of ATAC-seq reads at transcription start sites. Filtering for low quality nuclei and genes was applied separately for each sample using a custom function, code for which can be found on our GitHub repository[63] (see Code availability for more details). Doublet finder was used for doublet removal (v2.0.3) with a doublet score cut-off of 0.6. RNA-seq data was normalized using SCTransform with V2 regularization, while also regressing out cell cycle effects using cell cycle genes list available in the Seurat package. ATAC-seq data was normalized using TF-IDF normalization with default settings. Harmony (group.-by.vars='bio_origin') was then used to remove batch effects and integrate data from different samples, after which cells from both RNA-seq and ATAC-seq data were clustered using Weighted Nearest Neighbor (WNN) Analysis (FindMultiModalNeighbors() function with parameters reduction.list=list('SCT_CC_harmony', 'ATAC_harmony'); dims.list=list(1:30, 2:30); k.nn=30). Uniform Manifold Approximation and Projection (UMAP) was used to visualize the multiomics clusters (FindClusters() with setting Resolution=0.2).

## Differential expression analysis

For identification of differentially expressed genes between clusters, Wilcoxon's rank-sum test, implemented in the Seurat function FindAllMarkers() was used. Only genes expressed in at least 25% of cells were included in the analysis, and genes with log2 fold change >0.25 and FDR adjusted $p$-value < 0.05 were considered differentially expressed.

Differential gene expression between flight and ground conditions was performed using MAST, with a mixed model, using sample as a random effect. The analysis was done on each cell type separately, and included genes expressed in at least 30% of flight or ground cells in the given cell type. Genes with FDR adjusted $p$-value < 0.05 were considered differentially expressed. Differential chromatin accessibility between flight and ground conditions was performed the same way, using the same parameters.

## Gene overlap test

To test the significance of the overlapping genes found from the validation analysis, a hypergeometric distribution test was used to test the null hypothesis: The overlap of X genes between the two genelists is a random sampling effect. The phyper() function in R (v4.3.2) was used to perform the hypergeometric distribution test and the resultant $p$-value was recorded. The R scripts consisting of the code for this test is available on our github repository[63] (refer to Code availability for details).

## Motif analysis

ChromVAR (v1.16.0)[64], with default settings, was used to assess chromatin accessibility around DNA sequence motifs, taken from the

JASPAR (v2020_0.99.10) database[65]. For each motif and cell chromVAR calculated a score representing chromatin accessibility, while correcting for confounders such as for GC content and average accessibility. The chromatin accessibility scores from flight and ground cells were then compared, for each cell type separately. This was done with a mixed model using, sample as a random effect, implemented in the lmer() function from the lme4 (v1.1-28.9000) package. Motifs with FDR adjusted *p*-value < 0.05 were considered differentially accessible between flight and ground conditions.

## Gene and cluster annotation
Marker genes from the clustering analysis and the DEGs from the differential expression, for both ST and multiomics data, were using marker genes in the *Tabula muris* database[66], literature search and EnrichR pathway analysis tool (with mouse model organism used as background)[67–69]. These annotations for the marker genes were used as a guide for manual annotation of the multiomics as well as the ST clusters.

## Pathway analysis
For pathway analysis and visualization of the results, the Consensus Pathway Analysis platform[18] and the EnrichR pathway analysis tool[67–69] were used. For both these tools, pathway enrichment was determined relative to a universal background rather than a custom-defined, brain-specific gene list. This is to accept a known bias towards over-representation of brain-specific pathways in place of potential unknown biases resulting from the construction of a custom background gene list.

## Ligand-receptor analysis
Ligand-receptor interactions analysis was performed with the package CellPhoneDB (v3)[70]. Multiomics clusters were analyzed using the DE genes for spaceflight obtained from using MAST (v1.20.0) (see Differential Expression Analysis in Methods). For the analysis, DE genes were transformed to their human orthologues which is a required criteria to run CellPhoneDB.

The ligand-receptor analysis on the spatial data to identify co-expressed LR pairs was performed using SpatialDM (v0.1.0)[26]. The standard pipeline to perform global selection of significant LR pairs and for local selection of spots for co-expressed LR pairs was performed using the package documentation as a reference (with l = 250 and cutoff = 0.2 in sdm.weight_matrix() function). Likelihood ratio test was used in the differential testing and the test was run by following the package documentation. For further details on the scripts used in this analysis, refer to the Code availability section.

## Spatial pattern analysis
We inferred a gene regulatory network (GRN) from the brain multiomics dataset using CellOracle[43]. A tissue-specific GRN was generated by grouping all brain cell types into the same label before running the method. Transcription Factor (TF) activities were inferred for the brain spatial transcriptomics data using decoupler-py[41] with the method mlm and the obtained GRN as prior knowledge. Additionally, pathway activities were inferred for the brain spatial transcriptomics data using decoupler-py with the method mlm and the PROGENy model of pathway footprints as prior knowledge[42].

To analyze spatial relationships between cell types and pathway activities, we built cell type specific models with MISTy[40], which predict abundances (as determined by Stereoscope) from pathway activities in situ and from the local neighborhood (up to two spots away). In this case, The MISTy models are built only on Visium spots with at least 5% of the specific cell type.

We then extracted cell type-pathway interactions that occur in only one of the conditions. In the regions of interest where these differential interactions occur, we investigate changes in correlation (Pearson) between pathway activities and TF activities inferred from the CellOracle GRNs. We used Student's t-tests with Benjamini−Hochberg multiple testing correction for both determining the differential interactions and the changes in correlation.

## Metabolic pathway analysis
The flight-vs-ground expression level fold changes of each cluster were computed for genes that are expressed in more than 1% of the cells in either flight or ground condition of the cluster, using the FindMarkers() function in the Seurat package (v4.1.1)[71]. Genes listed in the RECON3D metabolic model[72] were extracted with their assigned metabolic pathway membership. The human gene IDs were translated to mouse gene IDs using the manual inspection and the HUGO Gene Nomenclature Committee (HGNC) Comparison of Orthology Predictions (HCOP) service[73]. All genes from the flight-vs-ground expression analysis were then ranked by fold-change differences (positive to negative) regardless of the *p*-value from FindMarkers() function, and tested for pathway enrichment vs the genes in RECON3D pathways using the fgsea function in the fgsea package (v1.22.0)[74] in R (v4.2.1)[75]. Resulting *p*-value enrichment of RECON3D-sourced pathways were adjusted (adjusted *p*-value) by program for the number of pathway tests performed.

## ShinyApp
ShinyApp was designed using the package shiny (v1.7.1) in R (v4.1.1) and the theme superhero was used for the overall esthetics of the shinyapp design. The R (v4.3.2) package babelgene (v22.9) with references from the NCBI database was used to identify the human orthologs for the genes listed under the Cluster markers - Expression and Spaceflight DEGs - Expression tabs in the shinyapp. All genes from Supplementary Data 4, 7 and 8 can be visualized on the shinyapp.

## RNAscope
Single molecule Fluorescence In Situ Hybridization (smFISH) to detect expression of genes of interest (Adcyl and Gpc55) was performed using the RNAscope technology following the manufacturer's protocol (Advanced Cell Diagnostics, ACD, Hayward, CA). Target genes (Gpc5 in channel red, Adcy1 in channel green) were visualized using the HRP-based RNAscope Fluorescent Multiplex Assay V2 (Cat. No. 323110). Brains from a cohort of 14−15 weeks-old female B6129SF2/J wild type mice flown to the ISS and housed in the Rodent Habitat for 28-29 days (NASA RR-10 mission) were used for this validation. Brains from mice of similar age, sex and strain were used as ground controls housed in identical hardware and matching ISS environmental conditions. At the end of the mission, spaceflight and ground controls were euthanized (euthanasia method: bilateral thoracotomy with sedation, ketamine/xylazine injection) and whole carcasses were stored at −80 °C until the dissection on earth. Brain tissues were harvested and snapped frozen in liquid nitrogen, then stored in −80 °C freezer for sample processing.

Brains from two ground controls (GC3, GC9) and three flights (FL1, FL3 and FL5) were cryo-sectioned to reach the hippocampus area and for each of them, one 10 µm section was placed on a superfrost plus slide and stored at −80 °C for less than a week before proceeding to the RNAscope protocol. The tissue sections were subjected to permeabilization steps before incubation with the target probes. The signal was amplified using the HRP-based technology (visualized by Tyramide Signal Amplification or TSA™ fluorophores by PerkinElmer).

## RNAscope signal quantification
The quantification of the fluorescence signal was performed with Fiji by ImageJ software. The images were taken with a ZEN LSM700 confocal microscope and exported as tiff with the same settings. Once exported they were loaded on ImageJ and the whole area of the section, i.e., the whole hemisphere was selected for quantification. Image was then transformed from RGB to 16-bit and quantification was done

by counting the number of pixels positive for the threshold set. The threshold was unique for each channel and was kept the same for ground and control samples.

## Reporting summary

Further information on research design is available in the Nature Portfolio Reporting Summary linked to this article.

## Data availability

The raw data generated in this study is hosted on the NASA GeneLab server, a part of the NASA Open Science Data Repository under study ID OSD-352[62] (https://doi.org/10.26030/jm59-zy54). This includes raw data (fastq files and brightfield images) from the ST and multiomics datasets, as well as processed data (aggregated gene count matrices, metadata, and fragment files) from the multiomics dataset. GeneLab, as an integral component of NASA's open-science initiative, provides researchers with raw and processed multi-omics data from spaceflight and ground-based analog experiments, fostering broad scientific collaboration[76,77]. Additionally, processed count matrices generated for the ST samples, the final data generated from the analysis performed in this study, and the source data for the generated figures are publicly accessible from Mendeley dataset[78] (https://doi.org/10.17632/fjxrcbh672.1). Furthermore, the RNAscope validation images with genes Adcy1 and Gpc5 for all the validated tissue sections are available on Figshare[79] (https://doi.org/10.6084/m9.figshare.24581544). Source data are provided with this paper.

## Code availability

All the necessary scripts required for the generation, processing, and analysis of the data discussed in this study are publicly available on our Github repository[63] (https://github.com/giacomellolab/NASA_RR3_Brain). Code to perform the validation tests and to generate the figures presented in this study are also available on this repository. Interactive visualization of our dataset can be done via our shiny app (https://giacomellolabst.shinyapps.io/rr3-brain-shiny/).

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

## Acknowledgements

We would like to gratefully acknowledge the assistance of San-huei Lai Polo with hosting data on the NASA GeneLab server. We thank Uppsala Multidisciplinary Center for Advanced Computational Science (UPPMAX) for providing computational infrastructure and Joakim Lundeberg for supporting Z.A. through Vetenskapsrådet funding. This study was supported by NASA GeneLab. GeneLab is funded by the Space Biology Program (Science Mission Directorate, Biological and Physical Sciences Division) of the National Aeronautics and Space Administration. S.G. was supported by Formas grant 2017-01066 and VR grant 2020-04864. The authors would also like to thank NASA Ames Research Center for providing the mouse samples to carry out this study.

## Author contributions

Conceptualization: S.G., J.M.G., S.V.C. Methodology: S.G. Investigation: Z.A. performed ST and single-cell multiomics experiments; V.B. performed sequencing, E.L. performed nuclei isolation. Formal analysis: E.C. performed data annotation, Z.A. and Å.B. performed ST data analysis, J.O.W. performed single-cell multiomics analysis, R.F. and P.BiM performed MISTy analysis, E.C., S.V. and D.T. performed pathway analysis with interpretations provided by R.G.H., C.S. and Y.M. performed Ligand-Receptor analysis, R.G.H. gave conceptual advice. Validation: R.F., P.BiM, D.C.W., M.G., S.F., K.M. Software: J.O.W., Å.B. Resources: V.B., A.S.B., D.C.W., J.M.G., S.V.C. Data curation: J.O.W., Å.B., A.S.B., C.S., R.F., Y.M., and P.BiM. Visualization: Y.M., Z.A., C.S. Supervision: S.G., O.B., J.S.R., J.M.G. Project administration: S.G., J.M.G., S.V.C. Funding acquisition: S.V.C., S.G. Writing – original draft: Y.M., E.C., Z.A., D.T., D.C.W. and S.G. Manuscript figures preparation: Y.M., Z.A. Supplementary Figs. and data preparation: Y.M., Z.A. Manuscript – review & editing: All authors.

## Funding

## Competing interests

ZA, SF and SG are scientific advisors to 10x Genomics Inc, which holds IP rights to the ST technology. SG holds 10X Genomics stocks. JSR reports funding from GSK, Pfizer and Sanofi and fees/honoraria from Travere Therapeutics, Stadapharm, Astex, Owkin, Pfizer and Grunenthal. All other authors declare no competing interests.

## Additional information

[1]Science for Life Laboratory, Department of Gene Technology, KTH Royal Institute of Technology, Stockholm, Sweden. [2]Space Biosciences Division, NASA Ames Research Center, Moffett Field, Mountain View, CA 94035, USA. [3]National Bioinformatics Infrastructure Sweden, Department of Biochemistry and Biophysics, Stockholm University, Science for Life Laboratory, Stockholm, Sweden. [4]Department of Cell and Molecular Biology, National Bioinformatics Infrastructure Sweden, Science for Life Laboratory, Uppsala University, Uppsala, Sweden. [5]Heidelberg University, Faculty of Medicine, and Heidelberg University Hospital, Institute for Computational Biomedicine, Bioquant, Heidelberg, Germany. [6]GSK, Cellzome, Heidelberg, Germany. [7]Bionetics, Yorktown, VA, USA. [8]Department of Biomedical and Health Informatics, The Children's Hospital of Philadelphia Research Institute, Philadelphia, PA, USA. [9]KBR, Space Biosciences Division, NASA Ames Research Center, Moffett Field, Mountain View, CA 94035, USA. [10]Department of Cell and Molecular Biology, Karolinska Institute, Stockholm, Sweden. [11]Department of Neuroscience, Karolinska Institutet, Biomedicum, Solna, Sweden. [12]NASA Postdoctoral Program - Oak Ridge Associated Universities, NASA Ames Research Center, Moffett Field, Mountain View, CA 94035, USA. [13]Pharmacology and Toxicology, Department of Pharmacology and Toxicology University Medical Center Goettingen, Goettingen, Germany. [14]Department of Pediatrics, The University of Pennsylvania Perelman School of Medicine, Philadelphia, PA 19104, USA. [15]Center for Mitochondrial and Epigenomic Medicine, Children's Hospital of Philadelphia and Department of Pediatrics, Division of Human Genetics, The University of Pennsylvania Perelman School of Medicine, Philadelphia, PA 19104, USA. [16]Department of Medicine, Karolinska Institute, Huddinge, Sweden. [17]These authors contributed equally: Yuvarani Masarapu, Egle Cekanaviciute, Zaneta Andrusivova. ✉e-mail: sylvain.v.costes@nasa.gov; stefania.giacomello@scilifelab.se

