## [Peer Review File · Nature Communications]

Spatially resolved multiomics on the neuronal effects induced by spaceflightREVIEWER COMMENTS

Reviewer #1 (Remarks to the Author):

In this report the authors combine single-cell multiomic (transcriptomics and chromatin accessibility) and spatial transcriptomics analyses and describe spaceflight-mediated changes in the mouse brain. This is the first on its kind study to use multiomic analysis in tissues from space flight. They compare ground control and spaceflight animals and found that the main processes affected by spaceflight include neurogenesis, synaptogenesis and synaptic transmission in cortex, hippocampus, striatum and neuroendocrine structures as well as astrocyte activation and immune dysfunction. However there are no validation or quantification for any of the target discovered.

The main alterations induced by spaceflight included changes in synaptogenesis, neuronal development as well as neurodegeneration associated with impaired protein folding and clearance, which overlapped with some of the effects of aging and neurodegenerative diseases. The results are interesting and recapitulate all the data previously reported in rodents and humans samples exposed to different space stressors. A major caveat is that there are no data validating the analysis they report.

A second major caveat is that the single nuc multiomics has a very low yield, ~3000 nuclei per samples they lose most of the immune cells. The clusters they identify primarily neurons and astrocytes and lack any other cell types from the brain this is a major caveat for the analysis and data interpretation that should be acknowledged.

snATAC-seq analysis in transcription factor binding sites indicated immune dysfunction in addition to spaceflight-mediated impairments of neurogenesis and synaptogenesis. In the sn ATC-seq they had a higher yield and collect also immune cells, they could not pick up the immune signal from the snseq because they did not get any immune cells. This demonstrate further that the low yeard lead to bias interpretation of the results.

Reviewer #2 (Remarks to the Author):

Masarapu and Cekanaviciute et al, investigated the impact of spaceflight on the central nervous system (CNS) in mice, aiming to identify potential health risks for astronauts during long-duration space missions. The study employs a novel approach by combining single-cell multiomics techniques, including transcriptomics and chromatin accessibility, along with spatial transcriptomics analyses to examine changes in the mouse brain caused by spaceflight.

By comparing ground control mice with those that experienced spaceflight, the researchers identified several major affected pathways by molecular phenotyping. Overall, the study provides novel insights on the spaceflight effects in the brain, however, the study needs to address important issues regarding the study design and data analysis:

1. The study employed whole hemisphere isolation for spatial data acquisition, as well as sections from the matching hemisphere. However, this approach may have resulted in an overrepresentation of certain cell types that were not present in the spatial data. I am particularly concerned about the prevalence of cells from the olfactory bulb in their dataset, considering the presence of a large cluster exhibiting neurogenesis in these young adult mice.
2. While the authors have appropriately highlighted clusters based on dominant biological processes, it is essential to include proper cell annotations for the main brain cell types. The absence of such annotations has caused the subsequent improper "cell-type" deconvolution in their analysis, which

relies on pathways rather than accurate cell type identifications.

3. The spatial data provided in the study exhibit non-matching brain sections, evident from the distinct bregma references. Unlike human samples, in mouse studies this is totally feasible, and it is important to ensure matching brain sections/axis to maintain precise brain orientation. Failure to use matching brain coordinates can introduce false discoveries during the differential gene expression analysis.

4. The ligand-receptor analysis performed in the study lacks spatial information. Given the solid tissue nature of the brain, it is crucial for the authors to incorporate spatial information to determine adjacent cell types and thereby increase the accuracy of this analysis, particularly in identifying potential ligand-receptor interactions.

5. The authors should provide clarification regarding the rationale behind the preference for using female mice and address any potential confounding effects of the estrus cycle in the experimental design given the age of the mice.

Minor

1. Can authors clarify whether the ground control and space flight were euthanized at (or near) the same time period and same way before the organs were harvested?

Reviewer #3 (Remarks to the Author):

In this manuscript, "Spatially resolved multiomics on the neuronal effects 1 induced by spaceflight," Masarapu, Cekanaviciute, and Andrusivova et al. used single-cell multiomics (10x multiome kits) and spatial transcriptomics (Visium) to map changes in the mouse brain (n=3) from spaceflight, compared to ground controls (n=3). They found neurogenesis, synaptogenesis, and synaptic transmission genes were affected in the cortex, hippocampus, striatum and neuroendocrine structures, with some additional evidence of astrocyte activation and immune dysfunction. Interestingly, the pathway analyses showed some indication of neurodegenerative diseases, plus oxidative stress and protein misfolding. Overall, they claim these changes and modified responses can help guide countermeasures.

The work is indeed some of the first of its kind, specifically the ligand-receptor interaction maps and results, and the authors are to be commended for also making a data visualization portal. Overall, I find the data interesting, worthy of publication, and which would be helpful for the field. However, I also had some questions and follow-up to their results that would help me make a final recommendation.

1) The authors only used the G1 ground control for the spatial data, but it's not clear why this was not also used for the snRNA-seq and snATAC-seq work. Could these data be added, or are the cells degraded or already used?

2) They found 825 differentially expressed genes (DEGs) induced by spaceflight across the sn-clusters, but how many of these overlap with known spaceflight DEGs? IF they compare to other GeneLab datasets, how many of these are novel vs. known?

3) Overall, there is little validation of the DEGs or omics results, such as with RT-PCR, orthogonal RNA-seq, or comparison with other data sets, and this would help the manuscript be much stronger if included.

4) For the TF enrichment, the methods are not clear, and this should be teased out more, such as if they used a tool like maxATAC or MEPP.

5) In general, there is a small methods section, and this should be expanded wherever possible. For

example, they state that "Data was then analyzed using the Seurat (v4.1.0) and Signac (v1.6.0) packages, but with which parameters? Which thresholds for the difference sections? This should be clarified.

6) Their shiny app only has Riken IDs for the labels, and they should add GeneIDs from standard nomenclature as well, and ideally label orthologs to human genes as well, which can help link to some of the disease states they mention.

7) The authors make a provocative statement that "anti-Parkinson's therapeutics might be repurposed as spaceflight countermeasures, or vice versa, that spaceflight could serve as an analog of accelerated aging-associated neurodegeneration" but this begs the question – which drugs? Based on which targets? It would be good to flesh this out in the discussion more.

8) Related to point #7, do the authors also observe signatures of aging, based on these data?

9) Since they have NGS data, did the authors look for any mutational signatures in the spaceflight samples? If they called variants on the data (not always accurate or easy on sc- or sn RNA-seq data), what would that look like?

Point-by-point response to Reviewers

We would like to thank the reviewers for their time in kindly reviewing our manuscript. We have addressed and responded to all of the reviewers' comments point-by-point below. We have revised the manuscript and have made multiple adjustments based on the comments, which we believe have further improved the manuscript.

Reviewer #1

Comments to authors

1. In this report the authors combine single-cell multiomic (transcriptomics and chromatin accessibility) and spatial transcriptomics analyses and describe spaceflight-mediated changes in the mouse brain. This is the first on its kind study to use multiomic analysis in tissues from space flight. They compare ground control and spaceflight animals and found that the main processes affected by spaceflight include neurogenesis, synaptogenesis and synaptic transmission in cortex, hippocampus, striatum and neuroendocrine structures as well as astrocyte activation and immune dysfunction. However there are no validation or quantification for any of the target discovered.

The main alterations induced by spaceflight included changes in synaptogenesis, neuronal development as well as neurodegeneration associated with impaired protein folding and clearance, which overlapped with some of the effects of aging and neurodegenerative diseases. The results are interesting and recapitulate all the data previously reported in rodents and humans samples exposed to different space stressors. A major caveat is that there are no data validating the analysis they report.

We thank the reviewer for acknowledging the novelty of our study. In order to validate our findings on the spaceflight affected processes in mouse brain, we performed single molecule Fluorescence InSitu Hybridization (smFISH) using the RNAscope technology for the precise detection and quantification of expression of two genes of interest (Adcy1 and Gpc5) in five brain sections: 3 flight, 2 ground controls from a comparative set of mice (**Supplementary Fig. 9**). Adcy1 is specific for its expression in the central nervous system (CNS)¹ and known to be associated with memory functions (synaptic plasticity and memory retention) in the brain², while Gpc5 is majorly expressed in the cortex, frontal lobe and cerebellum and is known to be involved in cell migration³. In our data, Adcy1 was particularly seen upregulated in the hippocampal CA3 (ST cluster 8) and dentate gyrus region (ST cluster 11) of the spaceflight samples (**Supplementary Table 8**). Adcy1 was also found to be upregulated in multiomics cluster 4 related to functions including neuronal activity and synaptic transmission. Gpc5 was found upregulated in astrocytes (multiomics cluster 4) of spaceflight samples (**Supplementary Table 3**). Both these genes have shown upregulation of expression in spaceflight samples compared to the ground controls in our data (**Supplementary Fig. 9A**).

Fluorescence signal for each gene from the RNAscope assay on the whole tissue sections was quantified to compare the flight and ground control sample groups (**Supplementary Fig. 9B, C**). This comparison revealed a significant upregulation in the expression of both genes in the spaceflight samples, confirming our findings from the ST data and multiomics data analysis. We have reported the results from the validation experiments in the ‘**Validation of spaceflight differentially expressed genes**’ section of the revised manuscript (**lines 368-382**).

For full transparency, the RNAscope experiment was conducted on mouse brains from a different mission (RR-10; see Methods “**RNAscope**” section) that were of comparable age to RR-

3 mice (14-15 weeks in RR-10 vs 12 weeks in RR-3), spent similar amount of time (28-29 days in RR-10 vs 39-42 days in RR-3) on the ISS and were euthanized using the same method (ketamine/xylazine injection). The reason for using these samples is the lack of remaining tissue from RR-3 mouse brains. In both missions, *Adcy1* and *Gpc5* showed significant increase in expression by spaceflight.

To extend our validation, we have also compared the spaceflight DEGs (Flight vs Ground Control; p-value < 0.05) with other spaceflight datasets available on NASA GeneLab. Spaceflight DEGs from bulk RNAseq analysis (629 DEGs; p-value < 0.05) of the brains from the same mice flown on the RR-3 mission showed an overlap of 11 genes with the 825 multiomics spaceflight DEGs found in our study. We also compared these 825 DEGs with other known spaceflight DEGs that were reported in a total of 11 other NASA GeneLab processed datasets which include mass spectrometry and RNAseq data collected from different organs of BALB/c and C57BL/6J mice strains. This comparison revealed 461 overlapping DEGs (p-value < 0.05) across all the 11 datasets combined (**Supplementary Table 5**). We have also reported these findings in the revised manuscript in lines **156-167, 415-420** and **496-500**.

2. A second major caveat is that the single nuc multiomics has a very low yield, ~3000 nuclei per samples they lose most of the immune cells. The clusters they identify primarily neurons and astrocytes and lack any other cell types from the brain this is a major caveat for the analysis and data interpretation that should be acknowledged.

We thank the reviewer for the comment and understand the concern about the low number of cells analyzed. To answer this comment, we want to emphasize that even if the number of cells is relatively low, we do not only have biological replicates for both spaceflight and ground control

samples but also we obtain two types of readouts (transcriptomics and chromatin accessibility) that allow us to obtain more robust cell clustering results. Based on that, in our dataset, we do identify one multiomics cluster specific for immune cells (multiomics cluster 10 annotated as Microglia) with a total of 319 cells. We identify immune marker genes *Ptprc* and *Cx3cr1* in this cluster in 127 and 232 cells, respectively, demonstrating that those are immune cells.

We would also like to emphasize that our study does primarily focus on neurons, astrocytes and oligodendrocytes. Most of the spaceflight-impairments related to immune dysfunction that we observe derive from the analysis on transcription factor (TF) activity in other non-immune multiomics clusters (i.e., multiomics clusters 11 and 14). Because of that, we have now added a sentence in the ‘**Conclusion**’ section of the revised manuscript (**lines 492-494**) to acknowledge this limitation as the reviewer suggested by stating that future studies to address specific immune dysfunction generated by spaceflight are needed.

3. snATAC-seq analysis in transcription factor binding sites indicated immune dysfunction in addition to spaceflight-mediated impairments of neurogenesis and synaptogenesis. In the sn ATC-seq they had a higher yield and collect also immune cells, they could not pick up the immune signal from the snseq because they did not get any immune cells. This demonstrate further that the low yeard lead to bias interpretation of the results.

We thank the reviewer for this comment. We understand that the use of the term ‘sn clusters’ interchangeably for the multiomics clusters in the manuscript may have caused some ambiguity. To correct this, we have now changed the terms ‘sn clusters’ to ‘**multiomics clusters**’ throughout the revised manuscript which we think is a better suited term referring to the resultant clusters from the integrated analysis of snRNAseq and snATACseq data in our study.

As stated in our reply for comment #2 above, we identified one multiomics cluster specific for immune cells (multiomics cluster 10 annotated as Microglia) with expression of immune marker genes like *Ptprc* and *Cx3cr1*. We identified most of the spaceflight-impairments related to immune dysfunction by analyzing the transcription factor (TF) activity in other non-immune multiomics clusters (i.e., multiomics clusters 11 and 14).

Reviewer #2

Masarapu and Cekanaviciute et al, investigated the impact of spaceflight on the central nervous system (CNS) in mice, aiming to identify potential health risks for astronauts during long-duration space missions. The study employs a novel approach by combining single-cell multiomics techniques, including transcriptomics and chromatin accessibility, along with spatial transcriptomics analyses to examine changes in the mouse brain caused by spaceflight.

By comparing ground control mice with those that experienced spaceflight, the researchers identified several major affected pathways by molecular phenotyping. Overall, the study provides novel insights on the spaceflight effects in the brain, however, the study needs to address important issues regarding the study design and data analysis:

Major comments

1. The study employed whole hemisphere isolation for spatial data acquisition, as well as sections from the matching hemisphere. However, this approach may have resulted in an overrepresentation of certain cell types that were not present in the spatial data. I am particularly concerned about the prevalence of cells from the olfactory bulb in their dataset, considering the presence of a large cluster exhibiting neurogenesis in these young adult mice.

This is a good point brought up by the reviewer. The combination of ST and multiomics used in this study is shown to be a powerful duo in extracting insights from the data and these two technologies complement each other. While the spatial data is a snapshot of only one coronal section, multiomics (snRNA-seq and snATAC-seq) can include more rostral and caudal to the tissue portion. For this reason, single cell omics (or multiomics) may pick up different cell populations, meaning that it can encompass regions not represented in the spatial data.

Regarding the multiomics clusters 0 and 9 which exhibit neurogenesis in our data, we compared their marker genes against region-specific elevated RNA⁴ which show genes (46 genes in total) with preferential expression in the olfactory bulb (OB). We could only find matches for two genes in multiomics cluster 0 (Pde1c, Neurod1) and none in cluster 9. Due to this, we believe that these clusters exhibiting neurogenesis are unlikely to include olfactory bulbs. However, since these clusters are not spatially coded, it does not rule out the case that these clusters contain neural progenitors (which are hard to distinguish by region) from the olfactory bulbs that do not yet have the OB-specific expression patterns.

In addition, anatomically there is a possibility that we have identified the neural progenitors in the olfactory bulbs during the process of migration from the subventricular zone via the rostral migratory stream.

2. While the authors have appropriately highlighted clusters based on dominant biological processes, it is essential to include proper cell annotations for the main brain cell types. The absence of such annotations has caused the subsequent improper "cell-type" deconvolution in their analysis, which relies on pathways rather than accurate cell type identifications.

Thank you for bringing this point regarding the cell annotations and the subsequent “cell-type” deconvolution from these annotations. We assigned cell annotations according to both major cell types as well as their functions. However, since we wanted to focus on the functions and processes involved in each cluster rather than just cell types, we made a conscious choice to include functions as cell annotations. We realized that it is beneficial to also include these cell type annotations for each cluster in the supplementary data as we had done in the case of ST clusters. Hence, we have now included an additional column ‘**celltype annotation**’ in ‘**Supplementary Table 2: Multiomics clusters annotations**’ that is provided along with the manuscript as additional files.

We used both functional and cell-type annotations for deconvolution and pathway analysis in our study. Specifically, we used the cluster ‘functions’ to identify functional similarities between the multiomics and ST clusters (**lines 227-238**) and ‘cell-type’ annotations to identify differential spatial patterns in the ST dataset (**lines 294-333**).

3. The spatial data provided in the study exhibit non-matching brain sections, evident from the distinct bregma references. Unlike human samples, in mouse studies this is totally feasible, and it is important to ensure matching brain sections/axis to maintain precise brain orientation. Failure to use matching brain coordinates can introduce false discoveries during the differential gene expression analysis.

We agree with the reviewer that non-matching brain coordinates can potentially introduce false discoveries during the differential expression analysis and that it is possible to obtain matching brain sections across mice. The main reason for not having precise matching brain sections across our samples is because the RR-3 brains were not preserved keeping in mind the

morphological aspect since, at the time of the sample dissection, brain analysis was not part of the study. In fact, after dissection, the brains were placed in 2 ml tubes and snap frozen causing the loss of some morphological information. Our study is the first of its kind on legacy spaceflight samples and besides providing new biological insights, it is a demonstration that by combining spatial transcriptomics and single-cell multiomics analyses, it is possible to extract meaningful information from the legacy samples present in the NASA Biological Institutional Scientific Collection (NBISC) repository.

That being said, since we became aware of the morphology issue at the start of our study, we first ensured that the RNA quality was not affected by the sample preservation approach by measuring RIN values for each sample and performing tissue optimization experiments to inspect the tissue morphology and RNA spatial distribution after sectioning as mentioned in the section **‘Spaceflight sample quality is suitable for ST and snMultiomics analysis’**.

To address inter-sample variability during differential expression analysis, we used MAST (v1.20.0) with a mixed model, using sample as a random effect as mentioned in the **‘Differential Expression Analysis’** section of **“Materials and Methods”**. This ensured that potential pseudoreplication bias or sample level differences were removed from the analysis. Moreover, we also ensured that only the clusters with comparable number of spots between all sections/samples were selected for interpretation of the results to avoid any unbalanced proportion of spots between tissue domains. To this end, for differential expression analysis we only considered the following spatial clusters: 8, 9, 14 and 16.

4. The ligand-receptor analysis performed in the study lacks spatial information. Given the solid tissue nature of the brain, it is crucial for the authors to incorporate spatial information to determine

adjacent cell types and thereby increase the accuracy of this analysis, particularly in identifying potential ligand-receptor interactions.

We thank the reviewer for the suggestion. The ligand-receptor interactions (LRIs) analysis was performed between the different spaceflight-affected multiomics clusters (or cell types) found in the multiomics data differential expression analysis. The distribution of the cells in relation to each other is not known for these clusters as we do not have spatial data exactly corresponding to these clusters. The same analysis on the ST clusters in our data, would still result in approximately 10-20 cells (same or of different cell types) in each spot, so would be unable to provide single-cell resolution. Due to this reason, we would not be able to correctly identify which cell/s in a spot are the source for the detected LRIs.

Hence, we would like to emphasize that though we agree with the reviewer that for single cell omics it would be helpful to incorporate spatial information of where each cell is located in relation to its neighbors, it is not possible using our techniques.

5. The authors should provide clarification regarding the rationale behind the preference for using female mice and address any potential confounding effects of the estrus cycle in the experimental design given the age of the mice.

We thank the reviewer for this valuable comment and we would like to shed light on the rationale behind the preference of using female mice in this study. The samples accessible to us are limited, because we can only request previously flown tissues, and there have been very few spaceflight studies on male mice due to the propensity to develop social behavior impairments when co-housed in a stressful environment.

Regarding the potential effects of the estrus cycle, this is a good point raised by the reviewer. Recent studies have shown that on the ISS the differences across individual male mice are greater than those across female mice⁵, thus indicating negligible effects of the hormonal changes occurring during the estrus cycle in female mice during spaceflight. We have also added this information in the **Methods** section under ‘**Animals**’ (lines 506-509).

Minor comments

1. Can authors clarify whether the ground control and space flight were euthanized at (or near) the same time period and same way before the organs were harvested?

Thank you for pointing this out. All mice (both ground controls and spaceflight) were euthanized in the same way using ketamine/xylazine/acepromazine. In addition to this, all mice were euthanized in the same week, between 17/05/2016 - 23/05/2016. In order to address this comment further, we have now added a sentence (lines 511-512) referencing the NASA OSD-352 study link in the ‘**Animals**’ section of ‘**Materials and Methods**’ that can be accessed to find the sample handling information for each individual mouse.

Reviewer #3

In this manuscript, “Spatially resolved multiomics on the neuronal effects 1 induced by spaceflight,” Masarapu, Cekanaviciute, and Andrusivova et al. used single-cell multiomics (10x multiome kits) and spatial transcriptomics (Visium) to map changes in the mouse brain (n=3) from spaceflight, compared to ground controls (n=3). They found neurogenesis, synaptogenesis, and synaptic transmission genes were affected in the cortex, hippocampus, striatum and neuroendocrine structures, with some additional evidence of astrocyte activation and immune

dysfunction. Interestingly, the pathway analyses showed some indication of neurodegenerative diseases, plus oxidative stress and protein misfolding. Overall, they claim these changes and modified responses can help guide countermeasures.

The work is indeed some of the first of its kind, specifically the ligand-receptor interaction maps and results, and the authors are to be commended for also making a data visualization portal. Overall, I find the data interesting, worthy of publication, and which would be helpful for the field. However, I also had some questions and follow-up to their results that would help me make a final recommendation.

Comments

1. The authors only used the G1 ground control for the spatial data, but it's not clear why this was not also used for the snRNA-seq and snATAC-seq work. Could these data be added, or are the cells degraded or already used?

We thank the reviewer for pointing this out. For our study, we split each brain into the two hemispheres, one for nuclei isolation and one for ST experiments (description in the '**Sample preparation**' of the **Methods** section in the revised manuscript). However, as the reviewer pointed out, we used one hemisphere from the G1 ground control mouse to optimize the nuclei isolation protocol. Consequently, we did not have enough G1 ground control brain material to prepare the multiomics (snRNA-seq and snATAC-seq) library for it. We have now added this information to the '**Nuclei isolation**' section in the '**Materials and Methods**' of the revised manuscript (**lines 556-557**).

2. They found 825 differentially expressed genes (DEGs) induced by spaceflight across the sn-clusters, but how many of these overlap with known spaceflight DEGs? IF they compare to other GeneLab datasets, how many of these are novel vs. known?

This is a very good suggestion. By comparing these 825 spaceflight multiomics DEGs to the 629 significant DEGs (Spaceflight vs Ground Control; p-value < 0.05) from the bulk RNAseq data of the same mice brains from the same NASA mission (RR-3), we found an overlap of 11 genes (**Supplementary Table 4**). Bulk RNAseq data reflects the overall abundance of a gene across the whole tissue/population of cells while the snMultiomics (snRNAseq specifically) reflects the gene expression at individual cell level. Due to this reason, we did not compare the genes based on their expression differences since these differences could produce contrasting logFC values in the resultant DE analysis. Interestingly, out of the 11 overlapping genes, only 2 genes (Gabra6, and Kctd16) were found to have the same direction of change in both the datasets indicating that the majority of spaceflight effects are cell type-specific and emphasizing the need for cell-specific analysis of central nervous system responses to spaceflight.

We also compared these 825 DEGs with known spaceflight DEGs that were reported in a total of 11 other datasets processed by NASA GeneLab (**Supplementary Table 5**). These datasets include mass spectrometry and RNAseq data collected from different organs of BALB/c and C57BL/6J mice strains. This comparison revealed a total of 461 overlapping DEGs (p-value < 0.05) across all the 11 datasets combined. For a detailed list of overlapping genes for each dataset, please refer to **Supplementary Table 5**. We have now reported these findings in the revised manuscript in lines **156-167, 415-420** and **496-500**.

3. Overall, there is little validation of the DEGs or omics results, such as with RT-PCR, orthogonal RNA-seq, or comparison with other data sets, and this would help the manuscript be much stronger if included.

This is an important point. In order to validate our findings on the spaceflight affected processes in mouse brain, we performed single molecule Fluorescence In situ Hybridization (smFISH) using the RNAscope technology for the precise detection and quantification of expression of two genes of interest (Adcy1 and Gpc55) in five brain sections: 3 flight, 2 ground controls from a comparative set of mice. Adcy1 is specific for its expression in the central nervous system (CNS)¹ and known to be associated with memory functions (synaptic plasticity and memory retention) in the brain², while Gpc5 is majorly expressed in the cortex, frontal lobe and cerebellum and is known to be involved in cell migration³.

Both these genes, in our study, have shown to be upregulated in spaceflight samples compared to the ground controls. Adcy1 was particularly seen upregulated in the hippocampal CA3 (ST cluster 8) and dentate gyrus region (ST cluster 11) of the spaceflight samples. Adcy1 was also found to be upregulated in multiomics cluster 4 related to functions like neuronal activity and synaptic transmission. Gpc5 was found upregulated in astrocytes (multiomics cluster 4) of spaceflight samples.

Quantitative analysis was done on the fluorescence signal obtained from the RNAscope validation experiments. Quantified signal for the whole tissue sections for each gene was done to compare the flight and ground control sample groups (**Supplementary Fig. 9**). This comparison revealed a significant upregulation in the expression of both the genes in the spaceflight samples, confirming our findings from the ST data and multiomics data analysis.

For full transparency, the RNAscope experiment was conducted on mouse brains from a different mission (RR-10; see Methods “**RNAscope**” section) that were of comparable age to RR-3 mice (14-15 weeks in RR-10 *vs* 12 weeks in RR-3), spent similar amount of time (28-29 days in RR-10 *vs* 39-42 days in RR-3) on the ISS and were euthanized using the same method (ketamine/xylozine injection). The reason for using these samples is the lack of remaining tissue from RR-3 mouse brains. In both missions, *Adcy1* and *Gpc5* showed significant increase by spaceflight.

We further quantified the fluorescence signal for gene *Adcy1* within the hippocampal CA3 and dentate gyrus regions. We found an increased expression of *Adcy1* within these regions of the spaceflight samples compared to ground controls, validating our observations from the DE analysis in the ST data. We have reported the results from the validation experiments in the ‘**Validation of spaceflight differentially expressed genes**’ section of the revised manuscript (**lines 368-382**).

4. For the TF enrichment, the methods are not clear, and this should be teased out more, such as if they used a tool like maxATAC or MEPP.

We thank the reviewer for pointing this out. We have now added a detailed ‘**Motif analysis**’ section under ‘**Materials and Methods**’ (**lines 638-646**) of the revised manuscript in order to make the TF enrichment part of the analysis clearer.

5. In general, there is a small methods section, and this should be expanded wherever possible. For example, they state that “Data was then analyzed using the Seurat (v4.1.0) and Signac (v1.6.0) packages, but with which parameters? Which thresholds for the difference sections? This should be clarified.

Thank you for the suggestion to expand the Methods by adding specific package settings and parameters. We have now revised the **‘Materials and Methods’** section accordingly. Particularly for subsections **‘ST data analysis’**, **‘Multiomics data analysis’**, **‘Differential Expression Analysis’** and **‘ShinyApp’**, we have expanded them by additional information like which package versions and functions were used in the analysis steps, and also the corresponding versions, parameter settings and thresholds that were applied. We hope this clarifies the review comment.

6. Their shiny app only has Riken IDs for the labels, and they should add GeneIDs from standard nomenclature as well, and ideally label orthologs to human genes as well, which can help link to some of the disease states they mention.

Thank you for pointing this out. Apart from most genes which were labeled by standard gene symbols in the shinyapp, there was also a small group of genes that were labeled by their Riken IDs (**Supplementary Table 13 and 14**). These genes were not assigned names hence their cDNA sequences at the time were only known either by their NCBI Entrez or Riken IDs provided by Mouse Genome Informatics (MGI). We have now added gene symbols (i.e., from the MGI database) for the genes that have now been assigned standard symbols. Additionally, to facilitate the gene search further, we have also added NCBI Entrez that provides a link to the gene record on the NCBI database.

We also thank the reviewer for the suggestion to label human orthologs that can be linked to some diseases mentioned in the study. We have now added human orthologs (gene symbols provided by HUGO Gene Nomenclature Committee or HGNC) too wherever there was a top match found with the *Mus musculus* gene. We used the R package `babelgene` (v22.9) and the NCBI

database to identify the human orthologs (please refer to ‘ShinyApp’ part of the **Methods** section in the revised manuscript). This extra information (gene symbols, NCBI Entrez and human orthologs respectively) added appears as column names ‘NCBI_name’, ‘NCBI_geneID’ and ‘human_ortholog’ in the table that appears when a gene is selected in the shinyapp (please see **Additional Figure 1** below).

Additional Figure 1: Snapshot of the shinyapp with the additional columns ‘NCBI_name’, ‘NCBI_geneID’ and ‘human_ortholog’. These correspond to the Gene symbols, NCBI Entrez ID and human orthologs respectively.

7. The authors make a provocative statement that “anti-Parkinson’s therapeutics might be repurposed as spaceflight countermeasures, or vice versa, that spaceflight could serve as an analog of accelerated aging-associated neurodegeneration” but this begs the question – which drugs? Based on which targets? It would be good to flesh this out in the discussion more.

This is a very good point and we thank the reviewer for pointing this out. As a response to this comment, we have now added below points in the revised manuscript from **lines 455-460** under the '**Discussion**' section.

Novel therapies for Parkinson's disease such as antioxidant (individual or a cocktail of antioxidants) treatments⁶ and mitochondria-targeting pharmaceuticals^{7,8} may thus be evaluated as countermeasures for spaceflight with proper FDA approvals. Alternatively, focussing on spaceflight effects found in this study like astrogliosis, microglia dysfunction, neuroinflammation, mitochondrial dysfunction and oxidative stress, might be useful in better understanding the underlying causes of age-related neurodegeneration.

8. Related to point #7, do the authors also observe signatures of aging, based on these data?

As noted in the section '**Metabolic gene enrichment analysis**', we observed a few typical signatures of aging, such as increased oxidative stress (caused by deficit in oxidative phosphorylation as mentioned in **lines 360-363**) and astrogliosis (astrocytes dysfunction caused by reduced arachidonic acid metabolism in spaceflight, **lines 365-366**) as signatures of aging. Both these signatures are known to be associated with Parkinson's disease⁹ and Alzheimer's disease^{10,11,12}, respectively.

9. Since they have NGS data, did the authors look for any mutational signatures in the spaceflight samples? If they called variants on the data (not always accurate or easy on sc- or sn RNA-seq data), what would that look like?

We thank the reviewer for the valuable suggestion to leverage the sequencing data for identifying mutations in the spaceflight samples. We used the tool SComatic¹³ to detect single-

nucleotide mutations in the snRNAseq data of the brain samples. We found several variants, ranging from 5-120 variants per sample (please see **Additional Figure 2** below). Due to the varied sequencing depth and the differences in the cell count across the samples, quantification of the variants is challenging. We normalized the variants count to the number of callable bases in order to account for the variability in sequencing coverage and calculated the number of callable bases for each cell type in the snRNAseq data. However, the resulting variants' count was not consistent across cell types, and variants found only for 10 multiomics clusters in total (multiomics clusters 0, 1, 2, 3, 4, 5, 6, 8, 10 and 17). Out of these 10 clusters, normalized variants were found across all the 5 mice samples only for multiomics cluster 0, Neurogenesis I (please see **Additional Figure 3** below).

This variability in variants' count in the spaceflight samples is supported by previous findings showing that variants detection in single-cell sequencing data is highly limited due to the differences in gene expression across cell types, varied sequencing depth as well the presence of sequencing errors or artifacts^{13,14}. Additionally, attributing to the slow cell division in the brain, every mutation that potentially arises from space radiation will only be present in one or very small population of cells. This would make it challenging to detect variants confidently and separation from noise will be even harder.

We also tried SComatic to detect somatic mutations in the snATACseq data of the brain samples which resulted in detection of no variants. This finding can be attributed to an even lower coverage in snATACseq data compared to the snRNAseq data where in the former, there's typically 0 or 1 reads per region in each cell. This makes it extremely difficult to detect any mutations with a reasonable confidence level in the ATACseq data.

Because of the technical challenges described above, we prefer to not include these analyses in the manuscript and we have reported them here for the reviewer.

Additional Figure 2: Detected variants across snRNAseq data of all the five mice samples in the multiomics dataset.

Additional Figure 3: Number of callable bases (normalized variants count) across all snRNAseq mice samples in multiomics cluster 0 (Neurogenesis I).

References

1. Shiers, S., Elahi, H., Hennen, S. & Price, T. J. Evaluation of calcium-sensitive adenylyl cyclase AC1 and AC8 mRNA expression in the anterior cingulate cortex of mice with spared nerve injury neuropathy. *Neurobiology of Pain* **11**, (2022).
2. Wang, H., Ferguson, G. D., Pineda, V. V., Cundiff, P. E. & Storm, D. R. Overexpression of type-1 adenylyl cyclase in mouse forebrain enhances recognition memory and LTP. *Nat. Neurosci.* **7**, 635–642 (2004).
3. Gpc5 glypican 5 [Mus musculus (house mouse)] - Gene - NCBI.

<https://www.ncbi.nlm.nih.gov/gene/103978#gene-expression>.

4. GENSAT Brain Atlas of Gene Expression in EGFP, tdTomato, and Cre Transgenic Mice, and Heintz TRAP mice.
http://www.gensat.org/anatomy_showcase_structure.jsp?structure_id=1&mouseType=EGFP
P.
5. Hong, X. *et al.* Effects of spaceflight aboard the International Space Station on mouse estrous cycle and ovarian gene expression. *NPJ Microgravity* **7**, 11 (2021).
6. Filograna, R., Beltramini, M., Bubacco, L. & Bisaglia, M. Anti-Oxidants in Parkinson's Disease Therapy: A Critical Point of View. *Curr. Neuropharmacol.* **14**, 260 (2016).
7. Gao, X. Y., Yang, T., Gu, Y. & Sun, X. H. Mitochondrial Dysfunction in Parkinson's Disease: From Mechanistic Insights to Therapy. *Front. Aging Neurosci.* **14**, (2022).
8. Malpartida, A. B., Williamson, M., Narendra, D. P., Wade-Martins, R. & Ryan, B. J. Mitochondrial Dysfunction and Mitophagy in Parkinson's Disease: From Mechanism to Therapy. *Trends Biochem. Sci.* **46**, (2021).
9. Oxidative phosphorylation mediated pathogenesis of Parkinson's disease and its implication via Akt signaling. *Neurochem. Int.* **157**, 105344 (2022).
10. Phillips, E. C. *et al.* Astrocytes and neuroinflammation in Alzheimer's disease. *Biochem. Soc. Trans.* **42**, 1321–1325 (2014).
11. Nardin, P. *et al.* Peripheral Levels of AGEs and Astrocyte Alterations in the Hippocampus of STZ-Diabetic Rats. *Neurochem. Res.* **41**, 2006–2016 (2016).
12. Impaired neurovascular coupling in aging and Alzheimer's disease: Contribution of astrocyte dysfunction and endothelial impairment to cognitive decline. *Exp. Gerontol.* **94**, 52–58 (2017).

13. Muyas, F. *et al.* De novo detection of somatic mutations in high-throughput single-cell profiling data sets. *Nat. Biotechnol.* 1–10 (2023).
14. Dou, J. *et al.* Single-nucleotide variant calling in single-cell sequencing data with Monopogen. *Nat. Biotechnol.* 1–10 (2023).

REVIEWER COMMENTS

Reviewer #1 (Remarks to the Author):

The authors responded to all the concerns

Reviewer #2 (Remarks to the Author):

Major comments

1. The study employed whole hemisphere isolation for spatial data acquisition, as well as sections from the matching hemisphere. However, this approach may have resulted in an overrepresentation of certain cell types that were not present in the spatial data. I am particularly concerned about the prevalence of cells from the olfactory bulb in their dataset, considering the presence of a large cluster exhibiting neurogenesis in these young adult mice.

> This is a good point brought up by the reviewer. The combination of ST and multiomics used in this study is shown to be a powerful duo in extracting insights from the data and these two technologies complement each other. While the spatial data is a snapshot of only one coronal section, multiomics (snRNA-seq and snATAC-seq) can include more rostral and caudal to the tissue portion. For this reason, single cell omics (or multiomics) may pick up different cell populations, meaning that it can encompass regions not represented in the spatial data. Regarding the multiomics clusters 0 and 9 which exhibit neurogenesis in our data, we compared their marker genes against region-specific elevated RNA4 which show genes (46 genes in total) with preferential expression in the olfactory bulb (OB). We could only find matches for two genes in multiomics cluster 0 (Pde1c, Neurod1) and none in cluster 9. Due to this, we believe that these clusters exhibiting neurogenesis are unlikely to include olfactory bulbs. However, since these clusters are not spatially coded, it does not rule out the case that these clusters contain neural progenitors (which are hard to distinguish by region) from the olfactory bulbs that do not yet have the OB-specific expression patterns.

In addition, anatomically there is a possibility that we have identified the neural progenitors in the olfactory bulbs during the process of migration from the subventricular zone via the rostral migratory stream.

>> "In the manuscript, the statement at line 230 regarding the deconvolution analysis, which suggests functional similarities between multiomics and spatial data clusters, requires further clarification. It appears that the authors are making comparisons between ST clusters based on cell type annotations and functional annotations derived from multiomics data. It is essential to define more explicitly what 'similarity' means in this context. Additionally, the manuscript would benefit from a detailed explanation of the statistical methods employed to establish the degree of agreement between these two datasets."

2. While the authors have appropriately highlighted clusters based on dominant biological processes, it is essential to include proper cell annotations for the main brain cell types. The absence of such annotations has caused the subsequent improper "cell-type" deconvolution in their analysis, which relies on pathways rather than accurate cell type identifications.

> Thank you for bringing this point regarding the cell annotations and the subsequent "cell-type" deconvolution from these annotations. We assigned cell annotations according to both major cell types as well as their functions. However, since we wanted to focus on the functions and processes involved in each cluster rather than just cell types, we made a conscious choice to include functions as cell annotations. We realized that it is beneficial to also include these cell type annotations for each cluster in the supplementary data as we had done in the case of ST clusters.

Hence, we have now included an additional column 'celltype annotation' in 'Supplementary Table 2: Multiomics clusters annotations' that is provided along with the manuscript as additional files.

We used both functional and cell-type annotations for deconvolution and pathway analysis in our study. Specifically, we used the cluster 'functions' to identify functional similarities between the multiomics and ST clusters (lines 227-238) and 'cell-type' annotations to identify differential spatial patterns in the ST dataset (lines 294-333).

>> "The current formatting of Supplementary Table 2 is difficult to read, primarily due to its orientation. Rescaling the table to fit on a single page would enhance its clarity. Additionally, it would be beneficial to include UMAP cluster annotations, along with a split-view UMAP comparison between control and spaceflight conditions. To further aid in visualization and quantitative analysis, a complementary barplot illustrating cell type proportions for each group and across all five samples is recommended. This would provide a clearer representation of cell proportions between the two groups and among the replicates.

Furthermore, the manuscript lacks a detailed description of the methods used for both functional and cell type annotations of clusters. While the source of the marker genes is identified, the specific approach for cell type annotation (manual or automated tool) remains unclear. Similarly, if EnrichR was used for functional analysis, the methods employed should be detailed. A significant concern with using EnrichR is the inability to adjust the background for pathway enrichment analysis, which can lead to an overrepresentation of expected neuronal pathways due to the default background being all genes expressed in mice. Adjusting the background to include only genes expressed in the brain would yield a more accurate representation of pathways for each cluster."

3. The spatial data provided in the study exhibit non-matching brain sections, evident from the distinct bregma references. Unlike human samples, in mouse studies this is totally feasible, and it is important to ensure matching brain sections/axis to maintain precise brain orientation. Failure to use matching brain coordinates can introduce false discoveries during the differential gene expression analysis.

> We agree with the reviewer that non-matching brain coordinates can potentially introduce false discoveries during the differential expression analysis and that it is possible to obtain matching brain sections across mice. The main reason for not having precise matching brain sections across our samples is because the RR-3 brains were not preserved keeping in mind the morphological aspect since, at the time of the sample dissection, brain analysis was not part of the study. In fact, after dissection, the brains were placed in 2 ml tubes and snap frozen causing the loss of some morphological information. Our study is the first of its kind on legacy spaceflight samples and besides providing new biological insights, it is a demonstration that by combining spatial transcriptomics and single-cell multiomics analyses, it is possible to extract meaningful information from the legacy samples present in the NASA Biological Institutional Scientific Collection (NBISC) repository. That being said, since we became aware of the morphology issue at the start of our study,

we first ensured that the RNA quality was not affected by the sample preservation approach by measuring RIN values for each sample and performing tissue optimization experiments to inspect the tissue morphology and RNA spatial distribution after sectioning as mentioned in the section 'Spaceflight sample quality is suitable for ST and snMultiomics analysis'.

To address inter-sample variability during differential expression analysis, we used MAST (v1.20.0) with a mixed model, using sample as a random effect as mentioned in the 'Differential Expression Analysis' section of "Materials and Methods". This ensured that potential pseudoreplication bias or sample level differences were removed from the analysis. Moreover, we also ensured that only the clusters with comparable number of spots between all sections/samples were selected for interpretation of the results to avoid any unbalanced proportion of spots between tissue domains. To this end, for differential expression analysis we only considered the following

spatial clusters: 8, 9, 14 and 16.

>> "While acknowledging the novelty of this study in analyzing legacy spaceflight samples, it is imperative to recognize that novelty alone does not validate the biological insights presented. The study lacks essential elements such as appropriate experimental controls, batch corrections, sampling strategies, and normalizations, which are critical for making biological claims. A striking example is the variation in 'Genes per spot' and 'UMI per spot' across samples, as highlighted in both ShinyApp and Supplementary Figure 3. The low UMI counts, particularly in samples F2-1, F2-2, and most G samples (except G1_1), raise questions about whether these differences are attributable to spaceflight or technical errors. This concern extends to the downstream analysis, such as DEG or pathway analysis, where the identified markers may be more reflective of technical artifacts than biological changes due to spaceflight. For instance, the genes *Wfs1*, *Dkk3*, and *Prox1*, emphasized in Figure 3, seem to correlate more with UMI counts than with biological effects. The reviewer strongly recommends additional deep sequencing to ensure comparable gene coverage for differential analysis. Downsampling is discouraged due to the shallowness of the lowest coverage samples.

Furthermore, the discrepancies in cell numbers between two sample groups shown in the ShinyApp, particularly for ST cluster 14, suggest a possible mismatch in brain sections between the G and F groups, leading to potentially misleading conclusions. The authors should employ a coordinated framework for a fairer comparison (e.g., same brain plane) or focus on clusters with matched molecular and spatial features. This concern also applies to cluster 16 (misabeled as 6 in ShinyApp). The assertions regarding protein misfolding and its implications for diseases like Parkinson's and Alzheimer's are speculative and lack statistical and empirical backing, particularly given the small sample sizes and missing data in certain groups. Such claims could mislead both the scientific community and NASA, impacting future spaceflight considerations. The authors are urged to critically reassess their findings and ensure that claims are scientifically substantiated.

Lastly, for Figure 3F, it is crucial to display the cell type proportions between G and F samples across all clusters. In Figures 3C and G, the tissue types should be labeled as F or G for clarity."

4. The ligand-receptor analysis performed in the study lacks spatial information. Given the solid tissue nature of the brain, it is crucial for the authors to incorporate spatial information to determine adjacent cell types and thereby increase the accuracy of this analysis, particularly in identifying potential ligand-receptor interactions.

> We thank the reviewer for the suggestion. The ligand-receptor interactions (LRIs) analysis was performed between the different spaceflight-affected multiomics clusters (or cell types) found in the multiomics data differential expression analysis. The distribution of the cells in relation to each other is not known for these clusters as we do not have spatial data exactly corresponding to these clusters. The same analysis on the ST clusters in our data, would still result in approximately 10-20 cells (same or of different cell types) in each spot, so would be unable to provide single-cell resolution. Due to this reason, we would not be able to correctly identify which cell/s in a spot are the source for the detected LRIs.

Hence, we would like to emphasize that though we agree with the reviewer that for single cell omics it would be helpful to incorporate spatial information of where each cell is located in relation to its neighbors, it is not possible using our techniques.

>> Please refer to the following papers:

<https://www.nature.com/articles/s41467-023-39608-w>

<https://www.nature.com/articles/s41592-022-01728-4>

<https://genomebiology.biomedcentral.com/articles/10.1186/s13059-022-02783-y>

In addition, please provide spatial and multiomics data showing ligand receptor pairs as figures.

Additional comments on FISH validation

>> "To validate the comparison between the two groups in the study, it is crucial for the authors to provide stained images of the entire brain or at least one hemisphere. Presenting these images would serve two important purposes. First, it would confirm that the scale of the region used for image quantification is consistent across both groups. Second, it would help ensure that any observed differences in expression are not a result of selection bias in the region of interest. Such visual evidence is essential to demonstrate the comparability of the brain sections being analyzed and to reinforce the validity of the findings."

Reviewer #2 (Remarks on code availability):

I have reviewed the code which are standard Seurat command lines. I did not download the data and ran the code however I was able to navigate their data via the ShinyApp provided the authors. Many of the genes mentioned in the manuscript were missing in the Shinyapp.

Reviewer #3 (Remarks to the Author):

In this revised manuscript, "Spatially resolved multiomics on the neuronal effects induced by spaceflight," Masarapu, Cekanaviciute, and Andrusivova et al. updated their paper on single-cell multiomics (10x multiome kits) and spatial transcriptomics (Visium) to map changes in the mouse brain from spaceflight, compared to ground controls. They found neurogenesis, synaptogenesis, and synaptic transmission genes were affected in the cortex, hippocampus, striatum and neuroendocrine structures, with some additional evidence of astrocyte activation and immune dysfunction. Interestingly, the pathway analyses showed some indication of neurodegenerative diseases, plus oxidative stress and protein misfolding.

They have added smFISH and some key validation data, as well as made nice additions to their data portal, which are great to see and will make this more of a resource for the field.

The updated paper overall is improved, and I just have a few questions for clarification remaining.

- 1) The authors validated their DEGs with 629 significant DEGs (Spaceflight vs Ground Control; p-value < 0.05) from the bulk RNAseq data of the same mice brains from the same NASA mission (RR-3), and they found an overlap of 11 genes. But, how significant is this overlap, and what is the null distribution of this result? I would like to see a permutation test of the overlap, and see how often 11 genes would overlap across these data sets?
- 2) The same question I have in point #1 applies to the 461 overlapping DEGs they found in their 11 other GeneLab DEGs in Supplementary Table 5. How often would these occur by chance? Overall I think these results are compelling, but it would be good to know how often they occur in the same direction and to the same degree, either with an updated Supplemental table or other additional figure at the end.
- 3) Related to #1 and #2, if the known DEGs they have curated could also be placed on their data portal, that would be nice.

Thank you.

Reviewer #3 (Remarks on code availability):

Reviewed their data portal and plotting functions.

Point-by-point response to Reviewers

Reviewer #1 (Remarks to the Author):

1. The authors responded to all the concerns

We thank the reviewer for taking the time in reviewing our manuscript.

Reviewer #2 (Remarks to the Author):

Major comments

1. "In the manuscript, the statement at line 230 regarding the deconvolution analysis, which suggests functional similarities between multiomics and spatial data clusters, requires further clarification. It appears that the authors are making comparisons between ST clusters based on cell type annotations and functional annotations derived from multiomics data. It is essential to define more explicitly what 'similarity' means in this context. Additionally, the manuscript would benefit from a detailed explanation of the statistical methods employed to establish the degree of agreement between these two datasets."

We thank the reviewer for this comment and suggestion. The ‘similarity’ mentioned here in the deconvolution analysis refers to the similarities between the functional annotations assigned to each of the multiomics clusters and ST clusters which are also shown in **Supplementary Table 9 (lines 221-222)**. For the deconvolution analysis, the default Negative Binomial based model implementation of Stereoscope was used to calculate the celltype proportion probabilities associated with each spatial cluster after correcting for any potential biases due to library sizes and experimental techniques ¹. We have added details to the deconvolution analysis results in **lines 218-220** of the updated manuscript.

2. *"The current formatting of Supplementary Table 2 is difficult to read, primarily due to its orientation. Rescaling the table to fit on a single page would enhance its clarity. Additionally, it would be beneficial to include UMAP cluster annotations, along with a split-view UMAP comparison between control and spaceflight conditions. To further aid in visualization and quantitative analysis, a complementary barplot illustrating cell type proportions for each group and across all five samples is recommended. This would provide a clearer representation of cell proportions between the two groups and among the replicates.*

Furthermore, the manuscript lacks a detailed description of the methods used for both functional and cell type annotations of clusters. While the source of the marker genes is identified, the specific approach for cell type annotation (manual or automated tool) remains unclear.

Similarly, if EnrichR was used for functional analysis, the methods employed should be detailed. A significant concern with using EnrichR is the inability to adjust the background for pathway enrichment analysis, which can lead to an overrepresentation of expected neuronal pathways due to the default background being all genes expressed in mice. Adjusting the background to include only genes expressed in the brain would yield a more accurate representation of pathways for each cluster."

Thank you for the valuable suggestions. Supplementary Table 2 was likely automatically converted during the submission which might have altered its orientation. We are sorry for the inconvenience. We have now reformatted and rescaled all the supplementary tables and converted them into pdf documents to fix any potential orientation issues.

As suggested, to aid visualization, we have also added new split-view UMAPs for control and spaceflight conditions as **Supplementary Fig. 5** and **6** provided along with the manuscript. A barplot showing the cell type proportions between the two sample groups (flight and ground control) as well as across all five samples have been provided as **Supplementary Fig. 4B** and **4C** respectively. We refer to these new figures in the manuscript at **lines 220-221**.

Regarding the description of functional and cell type annotations, we have now added more details in the 'Gene and cluster annotation' of the Methods section at **lines 604-605**.

Regarding adjusted backgrounds and overrepresentation bias for pathway enrichment analysis (using both EnrichR and CPA), we would like to thank the reviewer for drawing attention to this salient concern in the interpretation of these analyses. This caveat is not unique to the analyses presented in this study and is a known limitation of pathway analysis as a general method. We recognize that the data presented here ought to be interpreted in full awareness of this limitation and have amended the text to call appropriate attention to this caveat (**lines 603, and 609-613**). We believe interpreting the analyses as presented (against a universal background) with awareness of the method's limitations is preferable to generating and running pathway enrichment against a custom background gene list specific to this study, which would come with its own interpretation caveats.

*3. "While acknowledging the novelty of this study in analyzing legacy spaceflight samples, it is imperative to recognize that novelty alone does not validate the biological insights presented. The study lacks essential elements such as appropriate experimental controls, batch corrections, sampling strategies, and normalizations, which are critical for making biological claims. A striking example is the variation in 'Genes per spot' and 'UMI per spot' across samples, as highlighted in both ShinyApp and Supplementary Figure 3. The low UMI counts, particularly in samples F2-1, F2-2, and most G samples (except G1_1), raise questions about whether these differences are attributable to spaceflight or technical errors. This concern extends to the downstream analysis, such as DEG or pathway analysis, where the identified markers may be more reflective of technical artifacts than biological changes due to spaceflight. For instance, the genes *Wfs1*, *Dkk3*, and *Prox1*, emphasized in Figure 3, seem to correlate more with UMI counts than with biological effects. The reviewer strongly recommends additional deep sequencing to ensure comparable gene coverage for differential analysis. Downsampling is discouraged due to the shallowness of the lowest coverage samples. Furthermore, the discrepancies in cell numbers between two sample groups shown in the ShinyApp, particularly for ST cluster 14, suggest a possible mismatch in brain sections between the G and F groups, leading to potentially misleading conclusions. The authors should employ a coordinated framework for a fairer comparison (e.g., same brain plane) or focus on clusters with matched molecular and spatial features. This concern also applies to cluster 16 (misabeled as 6 in ShinyApp). The assertions regarding protein misfolding and its implications for diseases like*

Parkinson's and Alzheimer's are speculative and lack statistical and empirical backing, particularly given the small sample sizes and missing data in certain groups. Such claims could mislead both the scientific community and NASA, impacting future spaceflight considerations. The authors are urged to critically reassess their findings and ensure that claims are scientifically substantiated.

Lastly, for Figure 3F, it is crucial to display the cell type proportions between G and F samples across all clusters. In Figures 3C and G, the tissue types should be labeled as F or G for clarity."

We would like to thank the reviewer for bringing up the concern with the low data yield as it is very important and has helped us refine the manuscript.

At the time of data generation for this project, we targeted the recommended 50k mean reads pairs per spot for the Visium fresh frozen assay. Because of the sample intrinsic variability, we employed stringent data analysis workflows to minimize technical biases and sample variations that could potentially affect our results.

Specifically, to address the variations in sequencing depth within and across samples, for both the ST and multiomics datasets, we first employed a normalization technique (i.e., SCTransform, in the clustering workflow, that uses Pearson residuals) at individual sample level (mouse in the case of multiomics data and each section in the case of ST data). Subsequently, we also applied an integration step when combining all the samples within each dataset (ST and multiomics) to preserve biological variations, i.e., the variations across sample conditions (flight and ground control), as discussed in the **lines 532-536** of the manuscript. We also applied Harmony (via `runHarmony()` function available in the Seurat workflow) to address technical batch effects (batches from handling different Visium capture areas in ST data; batches from different sample handling in the case of multiomics data). Additionally, we performed an extra integration step using Harmony in the multiomics data analysis pipeline to regress out confounding cell cycle effects. Detailed description of the steps and applied settings are mentioned in the “**ST data analysis**” (line 526-540) and “**Multiomics data analysis**” (line 547-568) sections of **Materials and Methods**.

In order to get a robust set of biologically significant spaceflight DEGs, we ran differential expression analysis using a mixed-effects model (using MAST v1.20.0; detailed description in **“Differential Expression Analysis”** section of the manuscript in **lines 570-580**). In this method, we modeled a fixed spaceflight condition with sample as a random effect ^{2,3}, thus making sure that the spaceflight DEGs are not a result of sample variations but instead are due to sample group condition (spaceflight vs ground control). We also experimentally validated genes *Adcy1* and *Gpc5* by RNAscope (**Supplementary Fig. 12, 13B-C**) for our previous version of the manuscript.

In addition to applying MAST for DGE analysis, to avoid any imbalanced results from potentially different proportions of spots and nuclei per cluster between flight and ground control conditions, we now present and discuss results deriving from an even smaller group of clusters as the reviewer suggested. Consequently, we have now removed the pathway figure panel from Fig. 3 corresponding to ST cluster 16 (shown as Fig. 3E in previous manuscript version) and removed the results on ST cluster 14 (from section **snMultiomics maps the effects of spaceflight on different cell types in spaceflown mouse brains**). We also changed figure panels 4A, B which now do not include LR pairs and motif accessibility differences, respectively, previously deriving from multiomics cluster 14 which has an uneven distribution of nuclei count between the two sample conditions (**lines 306-311**).

For the remaining presented results, we would like to highlight that our observations are in agreement between the ST and multiomics datasets. For example, our findings from the metabolic gene enrichment analysis showed reduced glycolysis and oxidative phosphorylation in several multiomics and ST clusters (**Supplementary Table 13, 14**), which are known to be associated with several previously reported mitochondrial impairments caused by spaceflight ⁴. We also found reduced arachidonic acid metabolism (reduced in both multiomics and ST datasets; **Supplementary Table 13, 14**), which is primarily produced by astrocytes and suggests astrocyte dysfunction as a potential target for future spaceflight CNS studies (**lines 323-332**). These observations are consistent with our findings from other analyses including DEG analysis

indicative of a good agreement between the spatial and multiomics datasets, and support the detection of protein misfolding.

Nevertheless, we fully agree with the reviewer that too strong assertions could mislead the scientific community and NASA. For this reason, we have now tuned down the text about how our findings resemble Parkinson's and Alzheimer's similarities. We have made necessary adjustments to the **Results**, **Discussion**, and **Conclusions** sections. In addition, and very importantly, in order for the reader to understand and be aware of the limitations with this data, we have also acknowledged the same in the **Discussion** section of the updated manuscript in the lines **399-410** and **422-424**.

Regarding the reviewer's comment on the figures, we would like to elaborate that Figure 3F displays the multiomics cluster proportions which have at least 10% presence in each ST cluster. We selected this threshold for the barplot because of the following reason: the presence of celltypes with very low proportions in the ST clusters (for instance, 0.005%) cluttered the barplot making it hard to see the signal from other celltypes which were present at larger proportions. We thank the reviewer for the suggestion, and hence for transparency, we have now also included the barplot displaying *all* the multiomics cluster proportions in **Supplementary Figure 4A** and added **lines 220** and **221** in the text to facilitate the reader.

As the reviewer also suggested, we have now included labels for flight and ground control ST sections in Figures 3C and 3F (previously 3G).

We have also corrected the labeling of cluster 16 (previously labeled cluster 6 in the individual clusters plots in the shinyapp) in the updated version of our shinyapp. We thank the reviewer for noticing this typo.

4. Please refer to the following papers:

<https://www.nature.com/articles/s41467-023-39608-w>

<https://www.nature.com/articles/s41592-022-01728-4>

<https://genomebiology.biomedcentral.com/articles/10.1186/s13059-022-02783-y>

In addition, please provide spatial and multiomics data showing ligand receptor pairs as figures.

Thank you for the suggestions. We extended the ligand-receptor (LR) analysis to our spatial dataset using SpatialDM ⁵ as suggested by the reviewer. We applied SpatialDM on each ST brain section to identify spatially co-expressing LR pairs and found a total of 1,260 LR pairs (**Supplementary Table 10**). The local interacting spots for one adhesion molecule LR pair as an example from this list is shown in **Supplementary Fig. 9**. We also performed differential testing (likelihood ratio test) for the observed 1,260 LR pairs between the two conditions (flight and ground control) and found a total of 134 differential LR pairs (differential p-value < 0.1; **Supplementary Table 11**). We have now updated the manuscript with these findings in **lines 247-253**.

5. "To validate the comparison between the two groups in the study, it is crucial for the authors to provide stained images of the entire brain or at least one hemisphere. Presenting these images would serve two important purposes. First, it would confirm that the scale of the region used for image quantification is consistent across both groups. Second, it would help ensure that any observed differences in expression are not a result of selection bias in the region of interest. Such visual evidence is essential to demonstrate the comparability of the brain sections being analyzed and to reinforce the validity of the findings."

Thank you for the comment and suggestions. We agree with the reviewer that providing the stained images of the entire brain hemispheres that were used for the validation experiments is indeed important. In fact, in our previous version of the manuscript, we provided the stained images for the entire five brain sections in our FigShare project, details are available in the **Data availability** section of the manuscript. Nevertheless, for completeness, we have now also added these stained images of the entire brain section for all the five samples to the manuscript as **Supplementary Fig. 12**.

Importantly, we would like to emphasize that the RNAscope signal quantification that we presented in our previous version of the manuscript (now shown in **Supplementary Fig. 13B**) derives from the RNAscope signal detected across the whole hemisphere for all the brain sections (FL1, FL3, FL5, GC3, GC9) for both validated genes (Adcyl and Gpc5) to avoid any selection bias as the reviewer pointed out. Thanks to the reviewer's comment we realized that this was not fully clear in our manuscript and to avoid any lack of clarity to the readers, we have now made changes to the legend of **Supplementary Fig. 13** stating that the signal quantification was performed on the entire brain section. Moreover, we added this specification to the section “**RNAscope signal quantification**” in Materials and Methods at **lines 689-690**.

6. I have reviewed the code which are standard Seurat command lines. I did not download the data and ran the code however I was able to navigate their data via the ShinyApp provided the authors. Many of the genes mentioned in the manuscript were missing in the Shinyapp.

We thank the reviewer for reviewing our code and the shinyapp. We have now updated the shinyapp to also include the 4,057 spaceflight DEGs (also listed in the **Supplementary Table 8** of the manuscript) which were missing in the previous version. The top 20 cluster markers from the ST data (**Supplementary Table 7**), and the 825 spaceflight DEGs from the multiomics data (**Supplementary Table 3**) are also presented in the shinyapp.

Reviewer #3 (Remarks to the Author):

In this revised manuscript, “Spatially resolved multiomics on the neuronal effects induced by spaceflight,” Masarapu, Cekanaviciute, and Andrusivova et al. updated their paper on single-cell multiomics (10x multiome kits) and spatial transcriptomics (Visium) to map changes in the mouse brain from spaceflight, compared to ground controls. They found neurogenesis, synaptogenesis, and synaptic transmission genes were affected in the cortex, hippocampus, striatum and neuroendocrine structures, with some additional evidence of astrocyte activation

and immune dysfunction. Interestingly, the pathway analyses showed some indication of neurodegenerative diseases, plus oxidative stress and protein misfolding.

They have added smFISH and some key validation data, as well as made nice additions to their data portal, which are great to see and will make this more of a resource for the field.

The updated paper overall is improved, and I just have a few questions for clarification remaining.

We are pleased to know that the reviewer finds our updated manuscript with the validation results useful and considers these additions as an overall improvement to the manuscript content.

1) The authors validated their DEGs with 629 significant DEGs (Spaceflight vs Ground Control; p -value < 0.05) from the bulk RNAseq data of the same mice brains from the same NASA mission (RR-3), and they found an overlap of 11 genes. But, how significant is this overlap, and what is the null distribution of this result? I would like to see a permutation test of the overlap, and see how often 11 genes would overlap across these data sets?

We thank the reviewer for the comment. To validate our 825 significant multiomics spaceflight DEGs with the known bulkRNAseq DEGs, we filtered the genes based on their p-value significance (< 0.05). We then performed a match between these lists and found an overlap of 11 genes. Though it can be argued that these p-value cutoffs can vary across different genelists depending on the sample size or the technique used, in this case, both the genelists are from the same mouse brains, which we believe makes the overlap quite significant.

We also performed a hypergeometric distribution test to test a null hypothesis that ‘the overlap of the 11 genes between the two genelists (genelist A being the multiomics significant DEGs and genelist B being the significant DEGs from bulkRNAseq data) is a random sampling effect’. We used the phyper() function in R to perform this test and got a p-value of 0.01582549 indicating a highly significant gene overlap confirming our findings. We have updated these findings and corresponding methods in **lines 147-148** and **lines 582-588** of the manuscript.

2) The same question I have in point #1 applies to the 461 overlapping DEGs they found in their 11 other GeneLab DEGs in Supplementary Table 5. How often would these occur by chance? Overall I think these results are compelling, but it would be good to know how often they occur in the same direction and to the same degree, either with an updated Supplemental table or other additional figure at the end.

Thank you for the comment. As in #1 above, we extended the same hypergeometric distribution test to the overlapping DEGs from the 11 GeneLab datasets mentioned in the manuscript. The resultant p-values for each dataset are added as an extra column (**'P-value from Hypergeometric distribution test'**) in **Supplementary Table 5**.

Though the observed p-values are significant in the case of most of these datasets, it would not be ideal to compare the direction of change for these genes as the data is from different mice tissue organs obtained through different technologies (RNAseq, mass spectrometry). These differences may produce varying gene expression differences and comparison of such datasets using logFC values as a criteria (i.e., direction of change) could be misleading. Therefore, to avoid such potential misleading results, we decided to not test these overlapping genes for their direction of change.

3) Related to #1 and #2, if the known DEGs they have curated could also be placed on their data portal, that would be nice.

We thank the reviewer for the suggestion. We would like to highlight that the 825 spaceflight DEGs (**line 140-141, Supplementary Table 3**) identified from the multiomics data in our manuscript are also presented in the shinyapp. These DEGs also consist of the 461 overlapping genes which were found from the comparison in our validation analysis (**line 153-158**). Presenting all of the curated known DEGs (from all the 11 genelab datasets) is not feasible to present on our shinyapp since the non-overlapping genes would not hit a match when looked up

in the multiomics dataset via our shinyapp. However, as an addition to the previous shinyapp version, we have now updated the same to also include all the genes mentioned in our manuscript (i.e., top 20 cluster markers from the spatial data as shown in **Supplementary Table 7**, and the 4,057 spaceflight DEGs from the spatial data in **Supplementary Table 8** as well as the 825 DEGs from the multiomics data shown in **Supplementary Table 4**) and this is stated in **line 664** of the updated manuscript to facilitate the reader.

References

1. Andersson, A. *et al.* Single-cell and spatial transcriptomics enables probabilistic inference of cell type topography. *Communications Biology* **3**, 1–8 (2020).
2. Finak, G. *et al.* MAST: a flexible statistical framework for assessing transcriptional changes and characterizing heterogeneity in single-cell RNA sequencing data. *Genome Biol.* **16**, 1–13 (2015).
3. Heumos, L. *et al.* Best practices for single-cell analysis across modalities. *Nat. Rev. Genet.* **24**, 550–572 (2023).
4. da Silveira, W. A. *et al.* Comprehensive Multi-omics Analysis Reveals Mitochondrial Stress as a Central Biological Hub for Spaceflight Impact. *Cell* **183**, (2020).
5. Li, Z., Wang, T., Liu, P. & Huang, Y. SpatialDM for rapid identification of spatially co-expressed ligand–receptor and revealing cell–cell communication patterns. *Nat. Commun.* **14**, 1–12 (2023).

REVIEWERS' COMMENTS

Reviewer #2 (Remarks to the Author):

This reviewer appreciates the response made by the authors. The revised version of the manuscript is much clearer and the conclusions are reasonable.